# Resurgent Na$^+$ currents promote ultrafast spiking in projection neurons that drive fine motor control

Benjamin M. Zemel[1], Alexander A. Nevue[2], Andre Dagostin [1], Peter V. Lovell[2], Claudio V. Mello [2✉] & Henrique von Gersdorff [1,3✉]

The underlying mechanisms that promote precise spiking in upper motor neurons controlling fine motor skills are not well understood. Here we report that projection neurons in the adult zebra finch song nucleus RA display robust high-frequency firing, ultra-narrow spike wave-forms, superfast Na$^+$ current inactivation kinetics, and large resurgent Na$^+$ currents (I$_{NaR}$). These properties of songbird pallial motor neurons closely resemble those of specialized large pyramidal neurons in mammalian primary motor cortex. They emerge during the early phases of song development in males, but not females, coinciding with a complete switch of Na+ channel subunit expression from Navβ3 to Navβ4. Dynamic clamping and dialysis of Navβ4's C-terminal peptide into juvenile RA neurons provide evidence that Navβ4, and its associated I$_{NaR}$, promote neuronal excitability. We thus propose that I$_{NaR}$ modulates the excitability of upper motor neurons that are required for the execution of fine motor skills.

[1] Vollum Institute, Oregon Health and Science University, Portland, OR 97239, USA. [2] Department of Behavioral Neuroscience, Oregon Health and Science University, Portland, OR 97239, USA. [3] Oregon Hearing Research Center, Oregon Health and Science University, Portland, OR 97239, USA. ✉email: melloc@ohsu.edu; vongersd@ohsu.edu

The speed and accuracy of muscle control and coordination depend on the spiking activity of upper motor neurons[1]. In mammals, a subclass of layer 5 pyramidal neurons (L5PNs) project from the motor cortex to various targets in the brainstem and spinal cord. These cells are involved in specific aspects of fine motor control and they produce narrow half-width action potentials (APs)[2,3]. Changes to their intrinsic properties have been implicated in facilitating the learning of complex motor skills in some species[4–8]. Notably, primates and cats possess varying numbers of very large L5PNs with wide-caliber myelinated axons, fast AP conduction velocities, and ultra-narrow AP spikes[9–11]. These specialized Betz-type cells, first discovered in the human motor cortex, send projections that often terminate directly onto lower motor neurons[7], and are thought to be involved in highly refined aspects of motor control[1,5,9,10].

Birds display a diverse array of complex behaviors and cognitive skills ranging from elaborate nest building and tool usage to episodic memory and vocal mimicry[12–14]. Remarkably, birds accomplish this without a typical six-layered neocortex, which underpins the capacity for complex motor skills in mammals[15,16]. Nonetheless, avian pallial nuclei form microcircuits that appear analogous to those in the mammalian neocortex[15–19]. In songbirds, the robust nucleus of the arcopallium (RA) plays a key role in singing and provides direct descending projections to brainstem motor neurons that innervate the avian vocal organ (syrinx; Fig. 1a)[20–22] and respiratory muscles[23]. RA projection neurons (RAPNs) can thus be considered analogous to L5PNs in the motor cortex[16–19]. Indeed, these cells share important features, like wide-caliber myelinated axons and multiple spine-studded basal dendrites[24], although RAPNs lack the large, multilayer-spanning apical tufted dendrites that are a hallmark of L5PNs[25–27]. However, detailed knowledge of the ion channel composition, biophysical properties, and firing patterns of RAPNs is limited. Therefore, it is still unclear to what extent they function in an analogous manner to L5PNs.

Zebra finch RAPNs face considerable spiking demands during singing, which requires superfast, temporally precise coordination of syringeal and respiratory musculature[28–30]. As the adult male sings, RAPNs exhibit remarkably precise spike timing (variance ~0.23 ms)[31]. RAPNs also exhibit increased burstiness during the developmental song learning period, their instantaneous firing rates changing from 100 to 200 Hz when they produce immature vocalizations (subsong) to 300–600 Hz when a song becomes mature (crystallized). Average overall spike rates of RAPNs increase from 36 Hz at subsong to 71 Hz in adults[32]. Song maturation thus correlates with reduced variability in the timing of increasingly high-frequency bursts with a refinement of single spike firing precision. Importantly, nerve firing rates of >75 Hz are required for force summation in the superfast syrinx muscles[33]. This high spike frequency in RAPNs during song production is energetically demanding as indicated by increased staining for cytochrome C in maturing males, but not female zebra finches[34].

Songbird RAPNs thus seem to share with Betz-type L5PNs similar evolutionary pressure for fast and precise signaling, a constraint that can lead to neurons in unrelated species sharing similar expression patterns for a specific repertoire of ion channels. A good example is electric fish species from different continents, which have convergently evolved electrocytes exhibiting similar physiology and molecular features[35,36]. Indeed, high-frequency firing electrocytes with extremely narrow AP spikes co-express fast activating voltage-gated $Na^+$(Nav) and $K^+$ (Kv) channels[37,38].

Here, we explored the excitability of zebra finch RAPNs, posing the following question: Are the properties of $Na^+$ currents in RAPNs specialized to produce narrow, non-adapting spikes that enable precise high-frequency burst firing? We provide compelling evidence linking the presence of the auxiliary Navβ4 subunits to resurgent $Na^+$ currents ($I_{NaR}$) in RAPNs. We also show that the ability of mature RAPNs to produce narrow APs strongly correlates with increasing Navβ4 expression during the early phases of song development in males, but not in females that do not sing. However, isolated male juveniles that do not learn a tutor's song still develop mature spiking properties in their RAPNs. We conclude that $I_{NaR}$ likely plays an important role in fine-tuning the intrinsic excitability of RAPNs, an observation that is likely relevant for understanding the physiology of L5PNs expressing Navβ4. Finally, we note that RAPNs share strikingly similar spiking properties and ion channel subunits with mammalian Betz-type L5PNs. Our data thus indicate that $I_{NaR}$ can play a fundamental role in shaping the excitability of cortical-like pallial areas of the vertebrate brain that are required for the execution of fast and precise motor skills.

## Results

**Spontaneous and evoked spiking in RAPNs.** Whole-cell current-clamp recordings (CC) performed in adult male zebra finch brain slices revealed that RAPNs fire spontaneously in the absence of synaptic input. AP spikes had large amplitudes and narrow, submillisecond durations with a prominent after-depolarization that facilitated the production of subsequent spikes (Fig. 1b–d; spikes at ~24 °C; see Table 1). Recorded RAPNs were easily distinguished from local inhibitory GABAergic interneurons based on differences in their spiking properties and morphology (Supplementary Fig. 1). These results are consistent with previous studies[24,39–42]. Phase plane plots from the first derivative of these spontaneous APs revealed a biphasic AP upstroke (Fig. 1e), which has not been previously described in songbirds. This biphasic AP is a hallmark of L5PNs in the mammalian neocortex and suggests a high density of voltage-gated $Na^+$ channels at the axon initial segment[43]. The phase plots also revealed large maximum rates of depolarization and repolarization (Fig. 1e; Table 1), which likely facilitates high-frequency firing during in vivo song production[32,44].

At physiological avian body temperature (~40 °C)[45] the spontaneous firing frequency increased four-fold compared to room temperature (Fig. 1b, f; Table 1)[46]. Importantly, the AP waveform had a smaller amplitude at 40 °C and an extremely narrow half-width (~0.18 ms; Fig. 1g; Table 1). The APs also exhibited faster maximum rates of depolarization and repolarization in the phase plot at 40 °C (Fig. 1h; Table 1). During positive current injections, adult RAPNs produced high-frequency AP trains at 40 °C (Fig. 1i). Large amplitude spikes were triggered for the duration of the current injection with only a modest initial adaptation of spike amplitude. In fact, RAPNs showed little change in the maximum rate of depolarization between the 1st and 2nd spikes, despite a faster instantaneous firing frequency (IFF) at 40 °C (Fig. 1j–k; for ~24 °C data see Supplementary Fig. 2). We also note that during negative current injections at room temperature, we observed a voltage sag that is consistent with HCN-mediated currents (5.0 ± 0.68 mV between the peak and steady-state voltage during a −150 pA step stimulus), as previously described for RAPNs[40].

Our data reveal narrower AP spikes in RA than those previously reported[24,39,40]. This may be due to previous studies using lower temperatures, neurobiotin cell fills[47], and/or higher external divalent ion concentrations[48]. These results show that RAPNs have similar electrophysiological properties to cortical L5PNs that also display APs with sub-millisecond half-widths, temperature-sensitive amplitudes, and modestly adapting AP firing during positive current injections[43,49–51]. Importantly, the

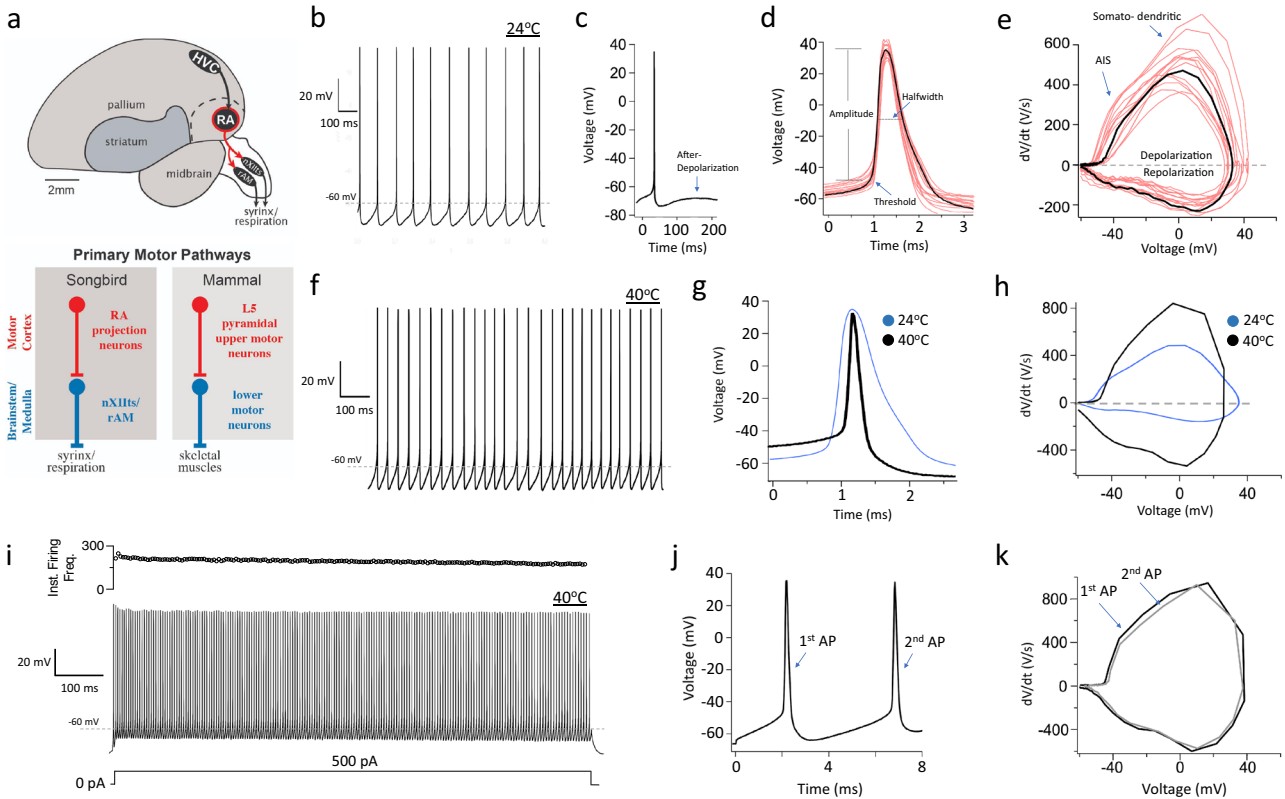

**Fig. 1 Ultra-narrow spikes and high frequency firing in RA projection neurons (RAPNs) of adult male zebra finches. a** Hyperdirect motor pathways in songbirds and mammals. Top: Zebra finch vocal control circuitry; nuclei and connections of the direct vocal-motor pathway shown in black, RA and its descending projections highlighted in red, other nuclei and connections omitted for clarity. The dashed line represents the dorsal boundary of the arcopallium. HVC is a proper name; RA, robust nucleus of the arcopallium; nXIIts, tracheosyringeal division of the hypoglossal nerve nucleus; RAm, nucleus retroambigualis medialis. Bottom: Diagram depicting upper (red) and lower (blue) motor neurons of the descending motor pathway in zebra finch, a songbird (left), and mammals (right). **b** Whole-cell current-clamp recording of spontaneous APs during 1 s at room temperature (24 °C). **c** Spontaneous AP from a different cell (same bird) than (**b**), arrow indicates after-depolarization. **d** Representative spontaneous AP from (**b**; black) with an overlay of spontaneous APs from an additional 15 RAPNs (pink); values for the indicated parameters are presented in Table 1. Data were obtained from multiple RAPNs from 6 adult male birds. **e** Phase plane plots of the AP in (**d**; black) and from another 15 RAPNs (pink, same cells as in **d**); indicated are the presumed contributions of the axonal initial segment (AIS) and somatodendritic components to the biphasic depolarization component in the phase plane plot. **f** Whole-cell current-clamp recording of spontaneous APs during 1 s in a RAPN at physiological temperature (40 °C). **g** Representative spontaneous AP recorded at physiological temperature (40 °C; black) with an overlay of a spontaneous AP recorded at room temperature (24 °C; blue). Recordings were taken from different birds in separate experiments. **h** Overlayed phase plane plots of the spontaneous APs from panel (**g**). **i** Representative AP train elicited by a 1 s 500 pA current injection at 40 °C; the corresponding plot of instantaneous firing frequency (Inst. Firing Freq.; in Hz) as a function of time is shown at the top. **j** First two APs from (**i**). **k** Overlay of the phase plane plots from the two APs in (**j**). The spike trains exhibit almost no adaptation.

prolonged high-frequency spiking of RAPNs suggests a high degree of availability of Nav channels. We, therefore, hypothesized that adult RAPNs contain a large pool of Nav channels and specialized mechanisms that limit their dwell time in nonconducting inactivated states.

**Navβ4 expression predicts a robust resurgent Na$^+$ current ($I_{NaR}$).** Recently, we have identified several transcripts for voltage-gated Na$^+$ channel beta (Navβ) auxiliary subunits as either positive (Navβ4) or negative (Navβ3) markers of adult male RA in the arcopallium[52]. While the functional role of these different Navβ subunits in RA neuronal excitability is unknown, Navβ4 is of particular interest because, in combination with Kv3 potassium currents[53], it can promote narrow spike waveforms and high-frequency firing via a resurgent Na$^+$ current ($I_{NaR}$)[54–56]. We, therefore, hypothesized that Navβ4 might be a key determinant of excitability properties in RA. High Navβ4 expression is restricted to RA in the arcopallium of adult male zebra finches[52], contrasting sharply with Navβ3, which is all but absent in RA (Fig. 2a, b). Navβ4 is thought to promote high-

frequency firing by limiting classical inactivation of Nav channels and promoting $I_{NaR}$[57]. Based on its expression levels, we predicted that neurons within RA would have a larger $I_{NaR}$ than those in the arcopallium outside RA. We thus performed wholecell voltage-clamp (VC) recordings at room temperature in RA sagittal slices from adult male finches (Fig. 2c). Voltage steps from −90 to +30 mV elicited large transient Na$^+$ currents ($I_{NaT}$) and, as predicted, subsequent steps to a range of test potentials (+15 to −75 mV) yielded robust $I_{NaR}$ (Fig. 2c) with a peak of −3.0 nA ± 0.28 (mean ± SEM) at the −45 mV test potential (Fig. 2e). The decay kinetics of $I_{NaR}$ was voltage-dependent, with single exponential decay time constants ranging from 2 to 45 ms (Supplementary Fig. 3a, b). These results reveal an exceptionally large $I_{NaR}$ in RAPNs compared to those recorded in cerebellum[56] and brainstem neurons[58].

In sharp contrast to RAPNs, neurons recorded in a caudal arcopallial region outside RA (Fig. 2a; shaded area), which has low Navβ4 expression, showed a much smaller $I_{NaR}$ (Fig. 2d, e). Notably, the recorded cells outside RA were morphologically distinct from neurons within RA, with relatively smaller somata

**Table 1 Passive and spontaneously active properties of male arcopallial neurons.**

| Age (dph) | Male ♂ | | | | | |
| --- | --- | --- | --- | --- | --- | --- |
| | **20** | **35** | **50** | **Adult RA (24 °C)** | **Adult RA (40 °C)** | **Adult (Outside)** |
| Input Res. (MΩ) | 422 ± 32.7[a,b,c] $N = 16$ | 285.4 ± 19.9[b,c,d] $N = 18$ | 175.7 ± 17.6[a,d] $N = 14$ | 190.2 ± 25.4[a,d] $N = 14$ | – | 276.0 ± 36.8 $N = 12$ |
| Tau (ms) | 39.0 ± 2.8[a,b,c] $N = 15$ | 28.0 ± 3.5[d] $N = 17$ | 25.9 ± 3.0[d] $N = 15$ | 18.8 ± 1.7[d] $N = 11$ | – | 47.5 ± 4.0[c] $N = 12$ |
| Capacitance (pF) | 103.3 ± 15.8 $N = 15$ | 109.0 ± 17.4 $N = 17$ | 155.6 ± 19.5 $N = 15$ | 114.3 ± 15.3 $N = 11$ | – | 195.6 ± 23.3[c] $N = 12$ |
| Spont. freq. (Hz) | 5.7 ± 0.7 $N = 14$ | 5.0 ± 1.4 $N = 17$ | 6.8 ± 1.0 $N = 17$ | 9.2 ± 2.5 $N = 17$ | 45 ± 10.3 $N = 8$ | 6.5 ± 2.5 $N = 4$ |
| % Spont. | 77.8 | 94.4 | 80.0 | 94.4 | 100 | 41.7 |
| Thresh. (mV) | −43.7 ± 0.8[c] $N = 13$ | −48.1 ± 1.5[c] $N = 16$ | −48.3 ± 0.7[c] $N = 17$ | −54 ± 2.0[a,b,d] $N = 14$ | −44.1 ± 0.7 $N = 8$ | −38.3 ± 0.4[c] $N = 4$ |
| HW (ms) | 1.8 ± 7E10−2[a,b,c] $N = 13$ | 1.2 ± 8E10−2[b,c,d] $N = 16$ | 0.8 ± 5E10−2[a,d] $N = 17$ | 0.6 ± 21E−2[a,d] $N = 14$ | 0.18 ± 0.02 $N = 8$ | 2.1 ± 0.2[c] $N = 4$ |
| Amp. (mV) | 85.1 ± 1.0 $N = 13$ | 88.6 ± 1.6 $N = 16$ | 90.0 ± 1.8 $N = 17$ | 90.5 ± 1.9 $N = 14$ | 62.7 ± 3.0 $N = 8$ | 62.3 ± 8[c] $N = 4$ |
| Max. D. (V/s) | 260 ± 15.1[a,b,c] $N = 13$ | 364.8 ± 20.6[b,c,d] $N = 16$ | 488.8 ± 22.1[a,d] $N = 17$ | 534.5 ± 33.6[a,d] $N = 14$ | 592.4 ± 66.5 $N = 8$ | 144 ± 30.8[c] $N = 4$ |
| Max. R. (V/s) | 42.8 ± 2.7[a,b,c] $N = 13$ | 77.9 ± 4.8[b,c,d] $N = 16$ | 130.4 ± 7.6[a,c,d] $N = 17$ | 181.9 ± 10.2[a,b,d] $N = 14$ | 478.9 ± 63.5 $N = 8$ | 41.1 ± 8.2[c] $N = 4$ |
| Estim. $I_{Na}$ (nA) | 26.9 | 39.8 | 76.1 | 61.1 | – | 28.2 |

A one-way ANOVA with a Tukey post hoc test was used for each measurement inside RA across the four ages (days post hatch, dph) indicated. Input Res.—input resistance, $P = 3.2 \times 10^{-10}$, $F_{(3, 63)} = 23.15$; Tau—membrane time constant, $P = 0.0003$, $F_{(3, 52)} = 7.494$; capacitance—membrane capacitance, $P = 0.12$, $F_{(3, 52)} = 2.013$; Spont. freq.—spontaneous firing frequency, $P = 0.26$, $F_{(3, 68)} = 1.379$; % Spont.—proportion of recorded cells that produced spontaneous APs, $P = 7.1 \times 10^{-10}$, $F_{(3, 63)} = 23.15$; Thresh.—AP threshold, $P = 4.5 \times 10^{-5}$, $F_{(3, 56)} = 9.281$; HW—AP half-width, $P = 4.5 \times 10^{-17}$, $F_{(3, 56)} = 57.31$; Amp.—AP amplitude, $P = 0.25$, $F_{(3, 56)} = 1.415$; Max. D.—maximum depolarization rate, $P = 1.5 \times 10^{-9}$, $F_{(3, 56)} = 22.03$; Max. R.—maximum repolarization rate, $P = 7.8 \times 10^{-18}$, $F_{(3, 56)} = 62.24$. Estim. $I_{Na}$—estimated Na$^+$ current during the depolarization phase of spontaneous APs as determined by the maximum repolarization rate × membrane capacitance. For comparisons between measurements within RA and outside RA in adults a two-tailed Student's t-test was used. Tau, $P = 2.1 \times 10^{-6}$; Capacitance, $P = 0.009$; Thresh., $P = 0.01$; HW, $P = 2.9 \times 10^{-5}$; Amp., $P = 0.004$; Max. D., $P = 0.0002$; Max. R., $P = 2.3 \times 10^{-5}$.
[a]Data are significant different from 35 dph ($P \leq 0.05$).
[b]Data are significant different from 50 dph ($P \leq 0.05$).
[c]Data are significant different from Adult RA ($P \leq 0.05$).
[d]Data are significantly different from 20 dph ($P \leq 0.05$).
P-values for significant post hoc comparisons:
Input resistance: 20 vs. 35 dph ($P = 0.0005$), 20 vs. 50 dph ($P = 1 \times 10^{-9}$), 20 vs. adult ($P = 5 \times 10^{-8}$), 35 vs. 50 dph ($P = 0.005$), adult vs. 35 dph ($P = 0.03$).
Tau: 20 vs. 35 dph ($P = 0.05$), 20 vs. 50 dph ($P = 0.01$), 20 vs. adult ($P = 0.0002$).
Half-width: 20 dph vs. adult ($P = 7.3 \times 10^{-12}$), 35 dph vs. adult ($P = 4.1 \times 10^{-7}$), 35 vs. 50 dph ($P = 0.0002$), 20 vs. 50 dph ($P = 7.4 \times 10^{-12}$), 20 vs. 35 ($P = 2.6 \times 10^{-7}$).
Maximum depolarization rate: 20 dph vs. adult ($P = 1 \times 10^{-8}$), 20 vs. 35 dph ($P = 0.02$), 20 vs. 50 dph ($2.9 \times 10^{-7}$), 35 vs. 50 dph ($P = 0.004$), 35 dph vs. adult ($P = 0.0001$).
Maximum repolarization rate: 20 dph vs. adult ($P = 7.3 \times 10^{-12}$), 20 vs. 35 dph ($P = 0.007$), 20 vs. 50 dph ($P = 1.4 \times 10^{-10}$), 35 vs. 50 dph ($P = 1.4 \times 10^{-5}$), 50 dph vs. adult ($P = 6.9 \times 10^{-5}$).

and dendrites with numerous, thin spines (Supplementary Fig. 1f). Because $I_{NaR}$ is a function of previously opened Nav channels[56], the expression of which can vary across cells, we normalized peak $I_{NaR}$ measurements to the peak $I_{NaT}$ (ratio of $I_{NaR}/I_{NaT}$). The average normalized peak $I_{NaR}$ was still much larger in RAPNs compared to neurons outside RA over a range of test potentials (−60 to −30 mV), with maximum normalized values (mean ± SE) of 0.28 ± 0.01 and 0.08 ± 0.01 at −45 mV for RAPNs and neurons outside RA, respectively (Fig. 2f). Decay kinetics also showed differences, with RAPNs exhibiting significantly smaller time constants at depolarized test potentials (−15 to +15 mV; Supplementary Fig. 3b).

Because Navβ4 has been linked to the facilitation of high-frequency firing in mammals and chicken[56,58], we predicted that RAPNs would be more excitable than neurons outside RA. Indeed, recordings performed at room temperature of spontaneous APs proved consistent with this prediction with RAPNs exhibiting narrower APs, larger AP amplitudes, and larger maximum rates of depolarization and repolarization than neurons outside RA (Table 1). Furthermore, while neurons both inside and outside RA were capable of increased spiking in response to current injections (Fig. 2g–i), RAPNs produced more spikes per second with a higher IFF for the first two APs at all positive current injections above +200 pA (Fig. 2i, j). These findings point to a strong correlation between Navβ4 expression, larger $I_{NaR}$, and greater intrinsic excitability, suggesting a role for Navβ4 in regulating intrinsic excitable properties of RAPNs in adult male zebra finches.

**Navβ4 and $I_{NaR}$ increase in parallel in RA during vocal development.** Male juvenile zebra finches progress through a critical period of vocal learning, during which vocal practice guided by auditory feedback is required to accurately produce a copy of the tutor song[12,59]. Zebra finches do not sing prior to ~28 days post hatch (dph). During the next phase (~28–45 dph) they produce unstructured vocalizations, referred to as subsong. They next enter a plastic phase (~45–90 dph), during which songs become more structured as the tutee refines syllable structure and sequencing guided by auditory feedback. By ~90 dph the song becomes crystallized and highly stereotyped. These developmental changes in song coincide with the growth of vocal control nuclei in males and changes in the connectivity of RAPNs[32,41,42,60–65]. In contrast, female finches do not develop a song and their song nuclei experience marked reductions in volume, as well as sharp decreases in neuronal cell number and size[61,62].

To investigate whether developmental changes in RA are also associated with changes in the expression of Navβ subunits, we performed in situ hybridizations for Navβ4 and Navβ3 in sagittal brain sections from male zebra finches at ages known to be within the pre-song (20 dph), subsong (35 dph), plastic song (50 dph), and crystallized song (>90 dph) stages of vocal development[59,66]. We observed an age-dependent increase in Navβ4 expression levels in RA, with significant differences between 20 and 35 dph juveniles compared to adults (Fig. 3a–d, i, black). We also detected an age-dependent increase in the proportion of Navβ4-expressing cells, progressing from 2% of cells relative to Nissl in

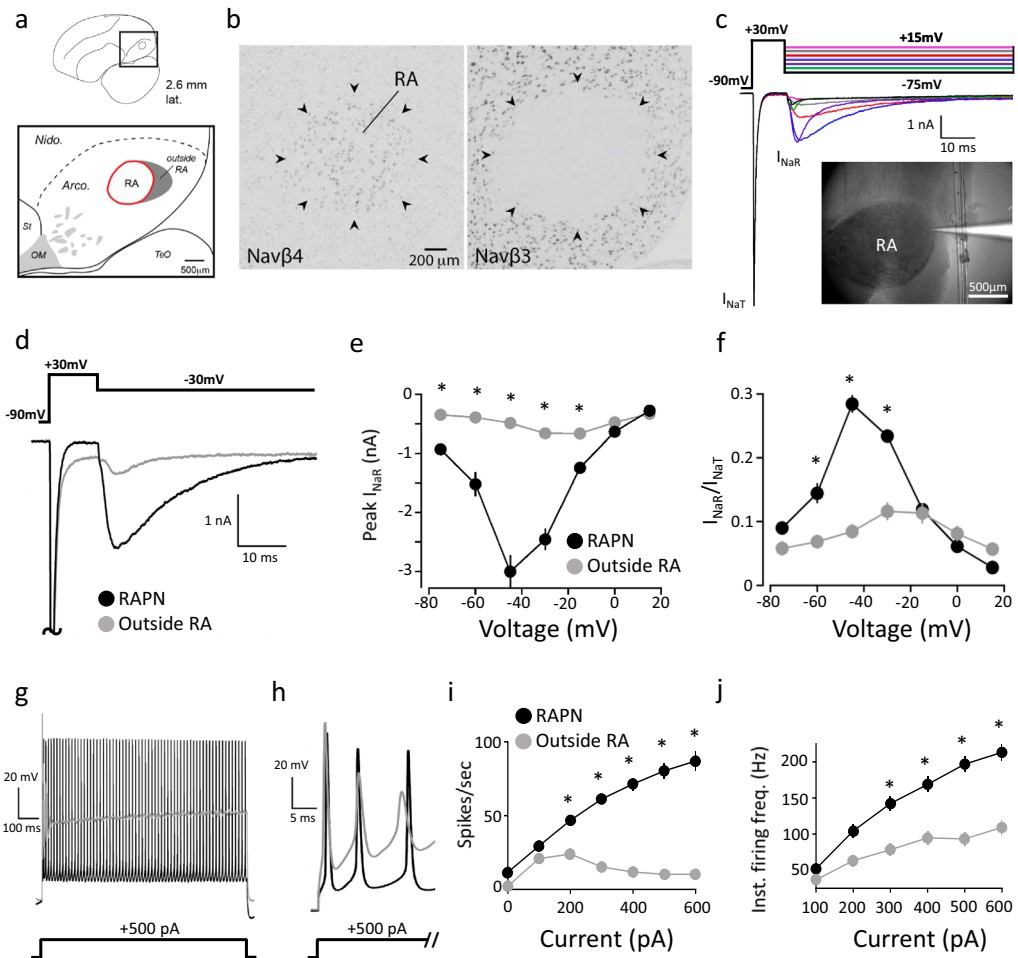

**Fig. 2 Expression of Navβ4 and β3 mRNAs, resurgent Na$^+$ current ($I_{NaR}$), and the spiking properties of RAPNs in the arcopallium of adult male zebra finches. a** Anatomical drawing indicating the position of RA within the arcopallium in a parasagittal section; distance from the midline indicated; the shaded area represents the region outside RA that is compared with recordings within RA. Arco., arcopallium; Nido., Nidopallium; OM, occipitomesencephalic tract; St, striatum; TeO, Optic tectum. Orientation: dorsal is up and anterior to the left. Adapted from Karten et al.[133]. **b** In situ hybridization images show high Navβ4 expression (left) and lack of Navβ3 expression (right) in RA (indicated by arrowheads). Each image is one of four replicates quantified in Fig. 3i–l. **c** Representative examples of $I_{NaT}$ and $I_{NaR}$ elicited in an RAPN at 24°C by the voltage clamp protocol at the top. $I_{NaT}$ was elicited via a 10 ms step to +30 mV followed by test potentials to +15 (pink), 0 (gray), −15 (red), −30 (blue), −45 (purple), −60 (green), and −75 mV (black) to elicit $I_{NaR}$. The cell was held at −90 mV during a 2 s intersweep interval. Inset: Image of a patch pipette filled with fluorescent dye for RA recordings. Quantified in 1e. **d** Detail views of example $I_{NaT}$ and $I_{NaR}$ elicited in an RAPN (black) and in a neuron outside RA (gray) by the voltage clamp protocol at the top. The $I_{NaT}$ peaks have been truncated. **e** Average I–V curves for the peak $I_{NaR}$ in RAPNs (black) and in neurons outside RA (gray) (two-way ANOVA with Tukey's post hoc; $P = 1.7 \times 10^{-21}$, $F (6, 146) = 27.00$, $N = 12$ RAPNs and 11 neurons outside RA; individual comparisons: −15 mV ($P = 0.009$), −30 mV ($P = 1.7 \times 10^{-18}$), −45 mV ($P = 3.0 \times 10^{-29}$), −60 mV ($P = 1.3 \times 10^{-8}$), −75 mV ($P = 0.008$)). Data are presented as mean values ± SEM. **f** Average I–V curves after normalization of the peak $I_{NaR}$ to the peak $I_{NaT}$ measured in a given sweep then averaged across cells (two-way ANOVA with Tukey's post hoc; $P = 6.1 \times 10^{-23}$, $F (6, 147) = 29.36$, $N = 12$ RAPNs and 11 neurons outside RA; Individual comparisons: −30 mV ($P = 1.1 \times 10^{-11}$), −45 mV ($P = 1.8 \times 10^{-26}$), −60 mV ($P = 1.3 \times 10^{-5}$)). Data are presented as mean values ± SEM. **g** Overlay of AP trains elicited by a 1 s 500 pA current injection in a RAPN and in a neuron outside RA at 24°C. **h** The first three APs from the AP trains shown in (**g**). **i** Average elicited spikes/s as a function of current injected (two-way ANOVA with Tukey's post hoc; $P = 2.5 \times 10^{-19}$, $F (6, 160) = 22.79$, $N = 14$ RAPNs, and 12 neurons outside RA; Individual comparisons: 200 pA ($P = 8.6 \times 10^{-4}$), 300 pA ($P = 2.6 \times 10^{-12}$), 400 pA ($P = 3.5 \times 10^{-19}$), 500 pA ($P = 2.7 \times 10^{-23}$), 600 pA ($P = 4.9 \times 10^{-25}$)). Data are presented as mean values ± SEM. **j** Average instantaneous firing frequencies measured as a function of current injected (two-way ANOVA with Tukey's post hoc; $P = 9.1 \times 10^{-5}$, $F (5, 138) = 5.647$, $N = 14$ RAPNs and 12 neurons outside RA; Individual comparisons: 300 pA ($P = 3.1 \times 10^{-4}$), 400 pA ($P = 3.5 \times 10^{-5}$), 500 pA ($P = 2.6 \times 10^{-10}$), 600 pA ($P = 7.4 \times 10^{-10}$)). Data are presented as mean values ± SEM. For data in **e**, **f** and **i**, **j**, * = $P \leq 0.05$ by post hoc analyses.

pre-song juveniles (20 dph) to 24%, 31%, and 63% in 35 dph, 50 dph, and adult birds, respectively, with a significant difference between adult and 20 dph finches (Fig. 3j, black). In contrast, no significant changes in Navβ4 expression levels, or the proportion of positive cells, were detected outside RA across ages (Fig. 3a–d, i, j, gray). In stark contrast to Navβ4, we found that Navβ3 expression levels and the proportion of Navβ3-expressing cells in RA decreased markedly across ages (Fig. 3e–h, k, l, black), with little evidence of change outside RA (Fig. 3e–h, k, l, gray). Finally,

we note that while Navβ2 is non-differentially expressed across the arcopallium, Navβ1 expression is higher in RA of adult males compared to the surrounding arcopallium[52]. We, therefore, asked whether Navβ1 is developmentally regulated. Similar to Navβ4, we found that Navβ1 expression was not present in nucleus RA in 20 dph males, contrasting markedly with the strong Navβ1 expression in adults (Supplementary Fig. 4).

Given the changes in Navβ4 expression, we predicted a corresponding age-dependent increase of $I_{NaR}$ in RAPNs. RA

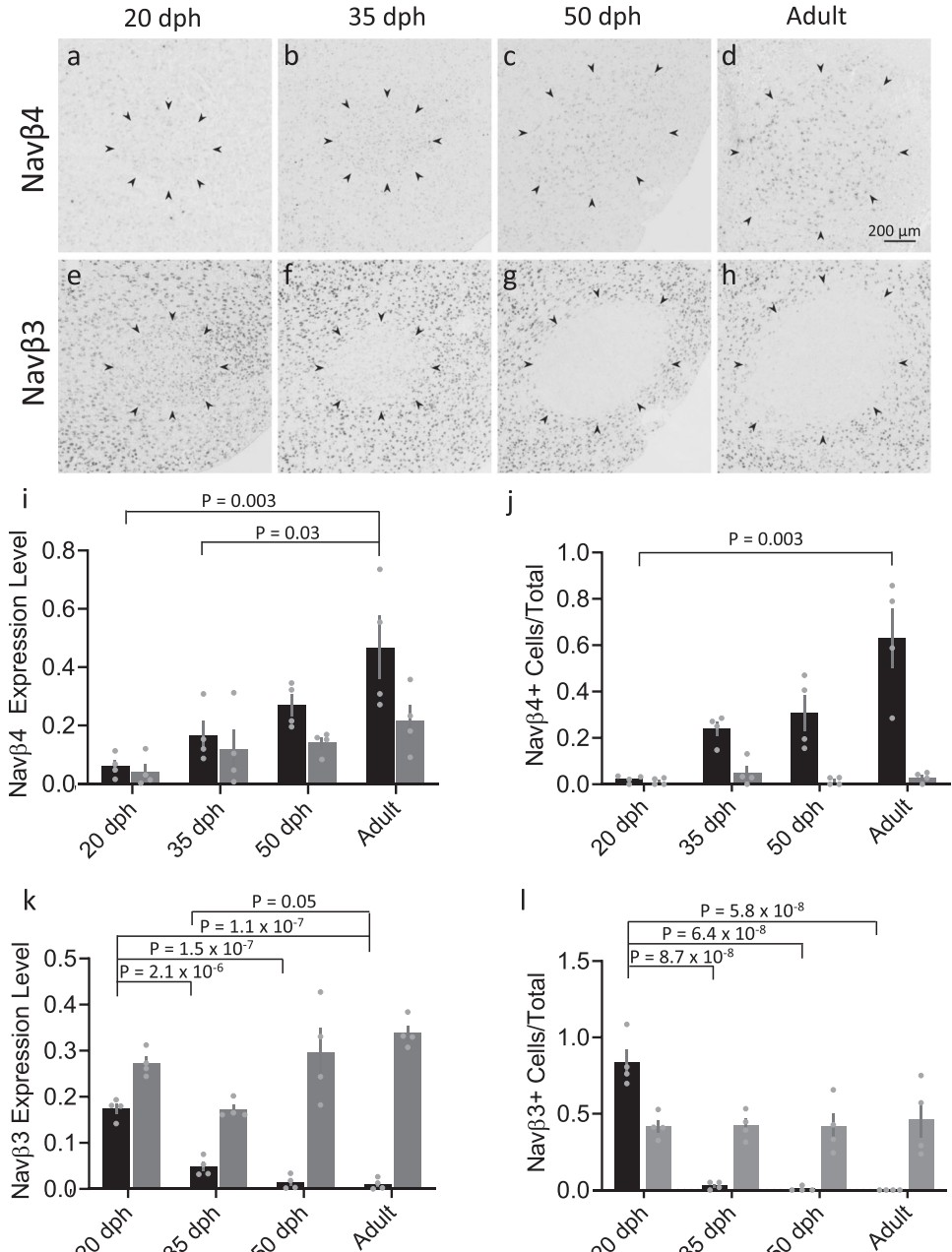

**Fig. 3 Age-dependent changes in expression of Navβ3 and Navβ4 mRNAs in the arcopallium of male zebra finches. a–d** Representative in situ hybridization images for Navβ4 mRNA within RA across ages indicated by days post hatch (dph); black arrowheads indicate RA borders. **e–h**. Representative in situ hybridization images for Navβ3 mRNA within RA across ages. Black arrowheads indicate RA borders. **i** Comparison of Navβ4 expression levels (normalized optical density) across age groups within RA (black) and in a caudal arcopallial region of equal size outside RA (gray). Significant age differences were observed in RA (one-way ANOVA with Tukey's post hoc; $P = 0.005$, $F (3, 12) = 7.416$, $N = 4$ males per age). Data are presented as individual data points with bars as mean values ± SEM. **j** Comparison of the proportions of Navβ4-expressing cells across age groups within RA (black) and in an arcopallial region of equal size outside RA (gray). Significant age differences in RA (one-way ANOVA with Tukey's post hoc; $P = 0.001$, $F = F (3, 12) = 10.79$, $N = 4$ males per group). Data are presented as individual data points with bars as mean values ± SEM. **k** Comparison of Navβ3 expression level (normalized optical density) across age groups within RA (black) and in an arcopallial region of equal size outside RA (gray). Significant age differences in RA (one-way ANOVA with Tukey's post hoc; $P = 5.4 \times 10^{-8}$, $F (3, 12) = 73.43$, $N = 4$ males per group). Data are presented as individual data points with bars as mean values ± SEM. **l** Comparison of the proportions of Navβ3-expressing cells across age groups within RA (black) and in an arcopallial region of equal size outside RA (gray). Significant age differences in RA (two-way ANOVA with Tukey's post hoc; $P = 1.6 \times 10^{-8}$, $F (3, 12) = 90.96$, $N = 4$ males per group). Data are presented as individual data points with bars as mean values ± SEM.

could be readily identified in sagittal brain sections obtained from 20 to 50 dph male finches via infrared differential interference microscopy (IR-DIC; Supplementary Fig. 5). Indeed, $I_{NaR}$ was small or absent in RAPNs from 20 dph finches, but its magnitude increased sharply with age (Fig. 4a, b), with significant age-

dependent effects observed for both peak $I_{NaR}$ (Fig. 4c) and normalized peak $I_{NaR}/I_{NaT}$ ratios (Fig. 4d). At the $-30$ mV test pulse, significant differences were found for post hoc pairwise comparisons of raw and normalized $I_{NaR}$ across all age groups. Importantly, the average normalized peak $I_{NaR}/I_{NaT}$ values were

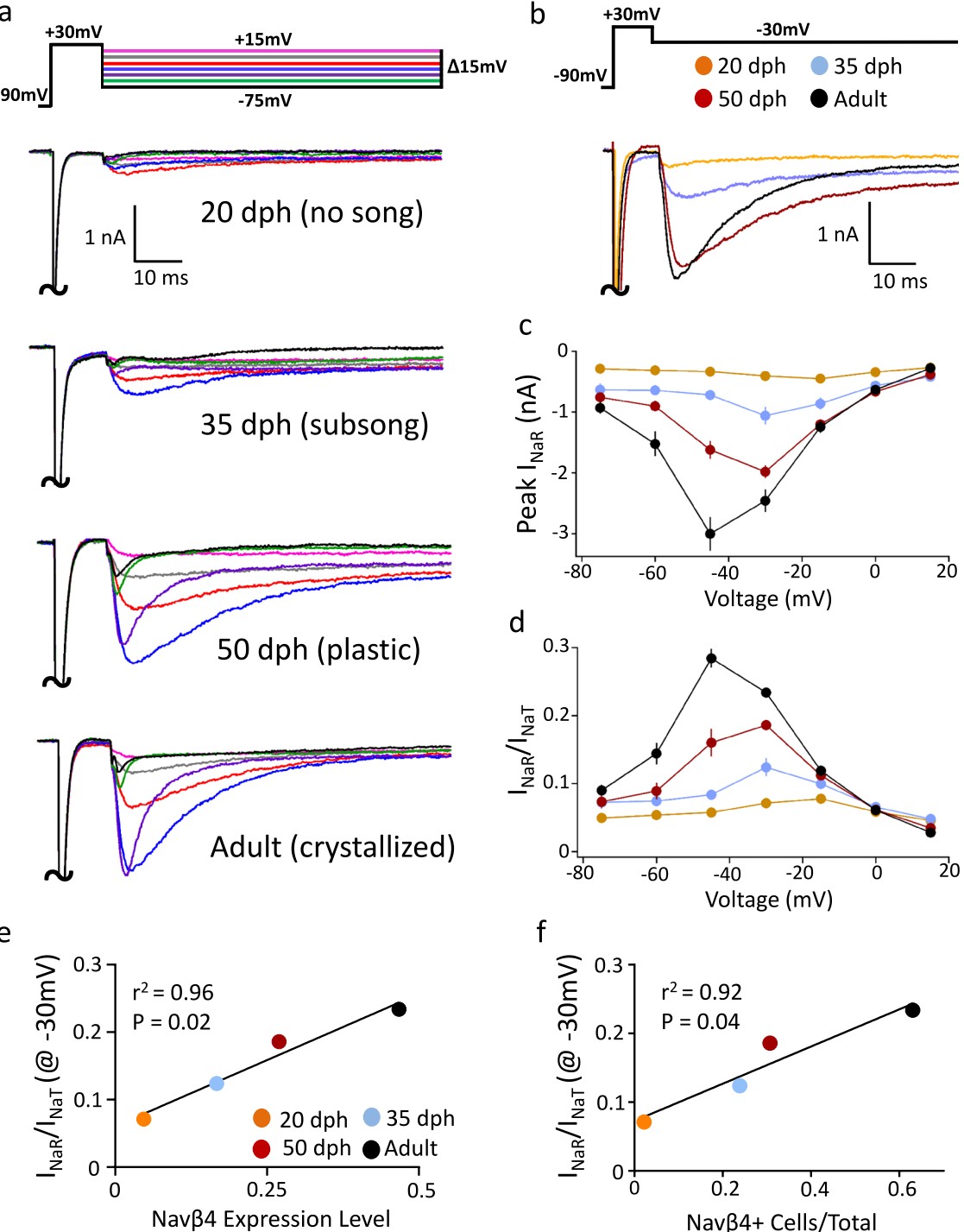

**Fig. 4 Age-dependent changes in $I_{NaR}$ in RAPNs of male zebra finches. a** Examples of transient ($I_{NaT}$) and resurgent ($I_{NaR}$) Na$^+$ currents elicited across ages (days post hatch, dph) by the voltage clamp protocol shown at the top. The large $I_{NaT}$ peaks have been truncated; the color code is the same as in Fig. 2c. **b** Representative currents from RAPNs at each age group during the −30 mV test potential shown at the top. The large $I_{NaT}$ peaks have been truncated. **c** Average I–V curves for the peak $I_{NaR}$ in RAPNs from each age group (two-way ANOVA with Tukey's post hoc comparisons; $P = 3.7 \times 10^{-35}$, $F_{(18, 280)} = 16.98$, $N$ (cells/age) = 10/20 dph, 12/35 dph, 10/50 dph and 12/adults). Data are presented as mean values ± SEM. **d** Average I–V curves after normalization of the peak $I_{NaR}$ to the peak $I_{NaT}$ measured in a given sweep then averaged across cells (two-way ANOVA with Tukey's post hoc comparisons; $P = 4.2 \times 10^{-46}$, $F_{(18, 280)} = 23.72$, $N$ (cells/age) = 10/20 dph, 12/35 dph, 10/50 dph, and 12/adults). Data are presented as mean values ± SEM. **e** Linear regression between the normalized average peak $I_{NaR}$ values at the −30 mV test potential across ages and the average Navβ4 expression level in RA (two-tailed; no adjustment for multiple comparisons). **f** Linear regression between the normalized average peak $I_{NaR}$ values at the −30 mV test potential across ages and the proportion of Navβ4-expressing cells in RA (two-tailed; no adjustment for multiple comparisons). For views of recorded slices across ages, see Supplementary Fig. 5a–c.

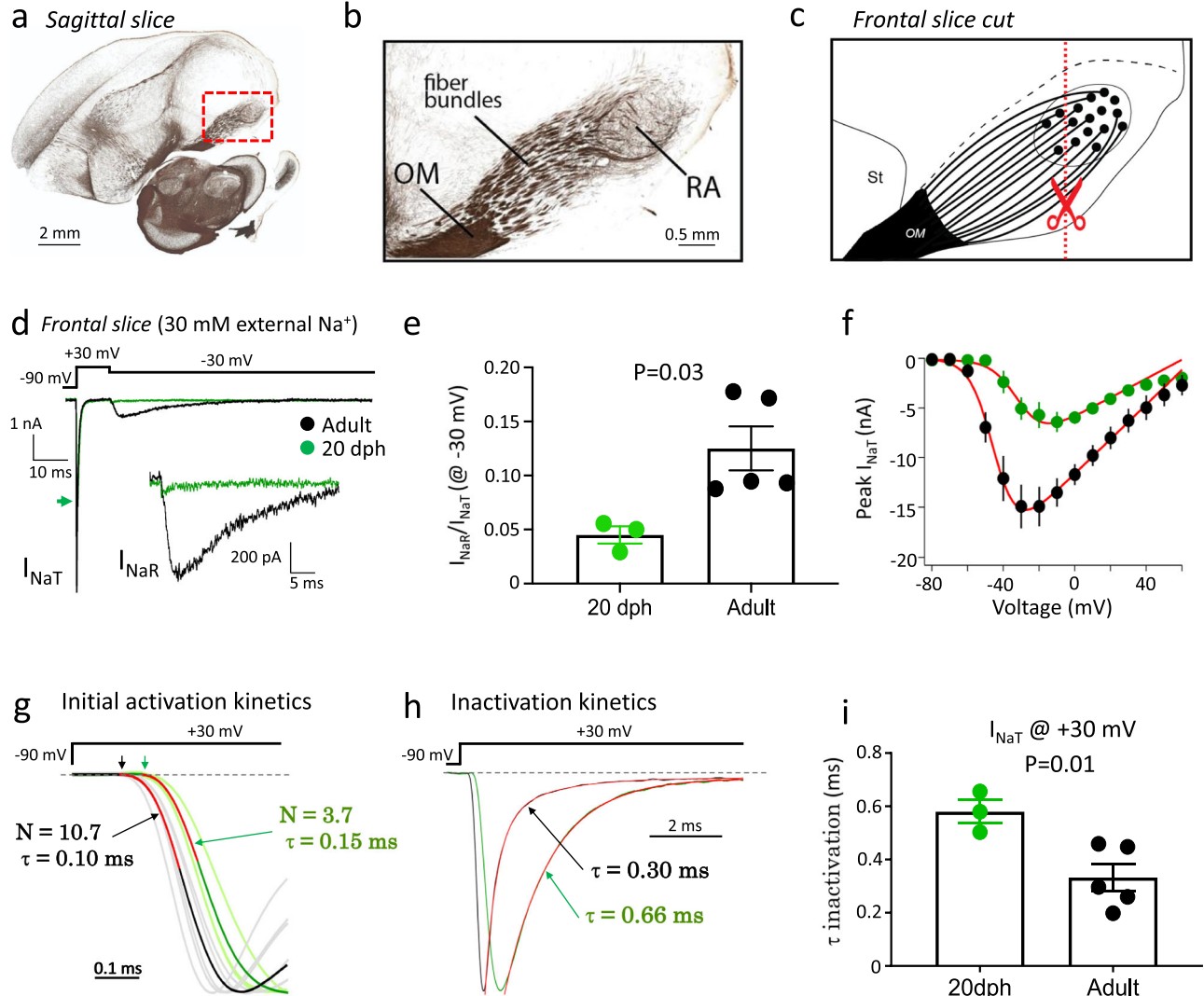

**Fig. 5 Recording $I_{NaT}$ and $I_{NaR}$ in RAPNs with reduced extracellular Na$^+$ in frontal slices from 20 days post hatch (dph) and adult male zebra finches at 24 °C. a** Myelin-stained parasagittal section containing nucleus RA (from zebrafinchatlas.org[128]; dorsal is up and anterior to the left). Adapted from Karten et al.[133]. **b** Detailed view of the area shown in the dashed red rectangle in (**a**) depicts myelinated fiber bundles that leave RA, run rostrally, and enter the OM (occipitomesencephalic tract). Adapted from Karten et al.[133]. **c** Schematic drawing depicts how generating frontal sections (e.g., dashed line with scissors) transects the axonal fibers closer to the somata of RAPNs (black dots in RA), resulting in shorter initial axonal segments of RAPNs than in parasagittal slices. **d** Overlay of representative transient ($I_{NaT}$) and resurgent ($I_{NaR}$) Na$^+$ currents from 20 dph (green) and adult (black) finches during a −30 mV test potential (top). The green arrow points to the 20 dph $I_{NaT}$ peak. $I_{NaR}$ has been enlarged in the inset. **e** $I_{NaR}/I_{NaT}$ measured at −30 mV in 20 dph and adult finches ($N$ (cells/age) = 3/20 dph and 5/adult finches; two-tailed Student's $t$ test). Data are presented as individual data points with bars as mean values ± SEM. **f** Average $I$–$V$ relationship for $I_{NaT}$ in 20 dph (green) and adult (black) finches ($N$ (cells/age) = 3/20 dph and 5/adult males). Data are presented as mean values ± SEM. Curve fits for the $I$–$V$ shown in the red lines. **g** Curve fits for the current activation shown (red) for averaged $I_{NaT}$ from 20 dph (green) and adult (black) finches where $N$ is the apparent number of transitions in a Markov chain model (equation shown in Methods), $\tau$ is the time constant of activation. Scaled $I_{NaT}$ from individual neurons used for averaging shown for adult (gray) and 20 dph (light green) finches. Arrows point to current onsets measured as the first point deviating from a line fit to the pre-stimulus baseline current. **h** Representative $I_{NaT}$ scaled to the peak from 20 dph and adult; the decay phases have been fitted with single exponentials (red) which overlay the respective traces. Note the small persistent Na$^+$ current under the dashed line. **i** Comparison of the time constant of decay for $I_{NaT}$ in 20 dph and adult finches ($N$ (cells/age) = 3/20 dph and 5/adult males; two-tailed Student's $t$ test). Data are presented as individual data points with bars as mean values ± SEM.

significantly correlated with both the average Navβ4 expression level (Fig. 4e) and the average proportion of Navβ4-expressing cells within RA (Fig. 4f). Therefore, Navβ4 expression strongly correlates with $I_{NaR}$ in RA and predicts the increase of $I_{NaR}$ seen in RA across different ages within the vocal learning period.

**Age-dependent changes in the activation and inactivation kinetics of $I_{NaT}$.** To address possible concerns that age-dependent differences in our VC recordings might be affected by space

and voltage clamp issues, we also performed recordings in frontal slices that transect RA axons near the soma (Fig. 5a–c). For those recordings, we lowered extracellular Na$^+$ from 119 to 30 mM to decrease the magnitude of $I_{Na}$ and compensated the series resistance electronically from 5 to 1 MΩ. These changes improve space and voltage clamping. Consistent with sagittal recordings with normal extracellular Na$^+$ (Fig. 4a–d), we observed a significant age-dependent increase in $I_{NaR}$ and confirmed that the normalized peak $I_{NaR}$ was significantly larger in the RAPNs of adults compared to 20 dph finches, where $I_{NaR}$ was almost

undetectable ($-30$ mV test pulse; Fig. 5d, e). We note that we are likely underestimating $I_{NaR}$ in these experiments as lowering external $Na^+$ ions disproportionately reduces $I_{NaR}$[56]. Importantly, the resulting $I–V$ plots of the peak $I_{NaT}$ from adults and 20 dph finches suggest that these currents are well voltage-clamped under these conditions (Fig. 5f)[67]. Adults had a larger $I_{NaT}$ amplitude that appeared to activate at more hyperpolarized potentials than in 20 dph juveniles (Fig. 5f). This likely leads to the faster AP upstroke and decreased threshold in adult RAPNs compared to juveniles (Table 1). In order to quantify the voltage dependence of $I_{NaT}$ we fit the averaged $I–V$ plots with the following Boltzmann function:

$$I = \frac{g_{max}(V - V_{rev})}{1 + \exp(\frac{V_{1/2}-V}{k})} \quad (1)$$

where $g_{max}$ is the maximum conductance, $V_{rev}$ is the reversal potential, $V_{1/2}$ is the half-maximal activation, and $k$ is the slope[68]. Fitting this function to the $I–V$ curves revealed that adults had a notably larger maximal conductance and leftward shifted half-maximal activation compared to juveniles. The values from the fits were as follows for adults and juveniles, respectively: $g_{max} = 174.0$ and $95.0$ nS; $V_{rev} = 66.5$ and $60.0$ mV, $V_{1/2} = -44.6$ and $-30.7$ mV, and $k = 6.6$ and $7.3$ mV. As predicted by the RAPN firing properties, the averaged $I_{NaT}$ also had a faster onset from the voltage step in adults (0.11 ms) compared to 20 dph juveniles (0.17 ms) (Fig. 5g). In order to quantify the kinetics of $I_{NaT}$ we fit the onset of this current with the following function:

$$I_{NaT} = I_{max}(1 - e^{-\frac{t}{\tau_a}})^N \quad (2)$$

where $\tau_a$ is the activation time constant and $N$ is the power-law exponent for the apparent number of transitions of $I_{NaT}$ activation as defined by a Markov chain model for a classic Hodgkin–Huxley $Na^+$ current activation m gate[69]. Interestingly, fitting this function from the onset of $I_{NaT}$ to ~40% of the peak during a $+30$ mV test pulse yielded an apparent number of activation gates in the averaged adult $I_{NaT}$ ($N = 10.7$) that greatly deviated from the Hodgkin–Huxley gating scheme ($N = 3$); this deviation was less apparent in the juvenile $I_{NaT}$ ($N = 3.7$; Fig. 5g). Similar fits to $I_{NaT}$ yielded an $N = 5$ for $+30$ mV steps from nucleated patches of L5PNs[69] and an $N = 9$ for $+10$ mV steps in cerebellar granule cells[70]. While fast inactivation may influence the accuracy of these measurements, they serve as a useful tool for quantitative comparisons between juveniles and adults. The time constants of activation were also smaller in adults ($\tau_a = 0.10$ ms) compared to juveniles ($\tau_a = 0.15$ ms) (Fig. 5g). Furthermore, the time constant of inactivation for $I_{NaT}$ was ~2-fold smaller in RAPNs from adults ($0.33 \pm 0.05$ ms; mean ± SEM) compared to 20 dph juveniles ($0.58 \pm 0.04$ ms) (Fig. 5h, i). This suggests the presence of a superfast Nav channel inactivation process that may be mediated, in part, by the fast block provided by Navβ4, as suggested in other systems[55,71]. During these studies, we also observed a small persistent $Na^+$ current in adult RA (Fig. 5h; $0.3 \pm 0.1$ nA at 100 ms after a voltage step from $-90$ to $-20$ mV at steady state).

**Age-dependent increases in the intrinsic excitability of male finch RAPNs.** We next investigated the intrinsic excitability of RAPNs during development using whole-cell CC. RAPNs showed spontaneous firing at all ages examined (Table 1; representative spontaneous spikes shown in Fig. 6a). The recorded APs exhibited significant age-dependent decreases in threshold and increases in maximum depolarization and repolarization rates (Fig. 6b, c; Table 1). We also found age-dependent changes in the passive properties, with significant decreases in both the input resistance ($R_{in}$), consistent with previous findings in male

finches[65], and in membrane time constant (Table 1). Although the calculated membrane capacitance ($C_m$) showed an upward trajectory across ages, we observed a trend toward a decrease from 50 dph to adults (Table 1). This observation mirrors those from a previous study[41]. This decrease in $C_m$ may be due to an increase in axonal compact myelination between these ages (Supplementary Fig. 5[72]). Furthermore, by multiplying the average $C_m$ (114 pF) by the maximum d$V$/d$t$ we calculated a large peak $Na^+$ current of 61 nA in adult RAPNs produced during a spontaneous AP (Table 1). By comparison, neurons of similar size ($C_m = 80$ pF) within deep cerebellar nuclei of mammals have only 21 nA currents[73]. These estimated peak $Na^+$ current values increased markedly across ages in finches (Table 1), a finding that is supported by VC experiments (Fig. 5d, f).

RAPNs at all ages examined were capable of increased AP firing in response to positive current injections (Fig. 6d–g, left). However, we observed a significant age-dependent increase in the number of spikes per second produced during current injections. APs in 20 dph finches began failing at current injections > +200 pA. Significant differences were seen when comparing 35 dph juveniles to either 50 dph or adults at current injections ≥ +400 pA (Fig. 6h). We also observed a significant age-dependent increase in the IFF, with significant differences when comparing 20 or 35 dph juveniles to either 50 dph or adults at multiple levels of injected current (Supplementary Fig. 6a). We thus conclude that both spike number and IFF reached a maximum at 50 dph that persisted through adulthood. Interestingly, although we observed a progressive increase in spike number produced with increasing age, the distribution of the membrane voltage ($V_m$) measured during the +500 pA current injection showed the opposite trajectory, with more time spent at hyperpolarized voltages for older ages (Fig. 6i). The overall average $V_m$ thus decreased with age (Fig. 6j). This increased residency at more hyperpolarized voltages in older finches is consistent with RAPNs that produce highly energy-efficient[74–77], narrow spikes that repolarize quickly in order to allow rapid recovery of $Na^+$ channels from inactivation and thus enable repetitive firing.

We next examined how features of AP waveforms changed during development. By overlaying the 1st, 2nd, and 16th–21st APs elicited by a +500 pA current injection, we were able to observe the effect of repetitive firing on spike amplitude and duration. This analysis revealed a progressive AP broadening and reduction in amplitude in 20 dph birds that began after the 1st AP, eventually reaching a steady state as the train progressed (Fig. 6d–g, middle). These effects were largely attenuated in older juveniles and adults (Fig. 6d–g, middle). Consistent with this finding, phase plane plots of 20 dph APs revealed a marked decrease in the maximum depolarization rate with increasing AP number, indicating a loss of Nav channel availability. This effect was noticeably less pronounced in older juveniles and adults (Fig. 6d–g, right). We also found a significant effect of age on the fractional loss in the maximum depolarization rate between the 1st and 2nd AP, with significant differences across all pairwise comparisons during the +600 pA current injection (Supplementary Fig. 6b). Together, these findings are consistent with the hypothesis that a large $I_{NaR}$ component is important for preserving Nav channel availability during high-frequency firing. These findings also provide evidence that the intrinsic excitability of RAPNs undergoes substantial developmental changes, in contrast with previous studies suggesting a lack of marked age-dependent changes[41].

To facilitate long periods of whole-cell recordings we routinely performed these CC experiments at room temperature (~24 °C) because recordings did not last long at higher temperatures. However, the gating kinetics of ion channels are highly dependent on temperature, with $Q_{10}$ values varying widely across different

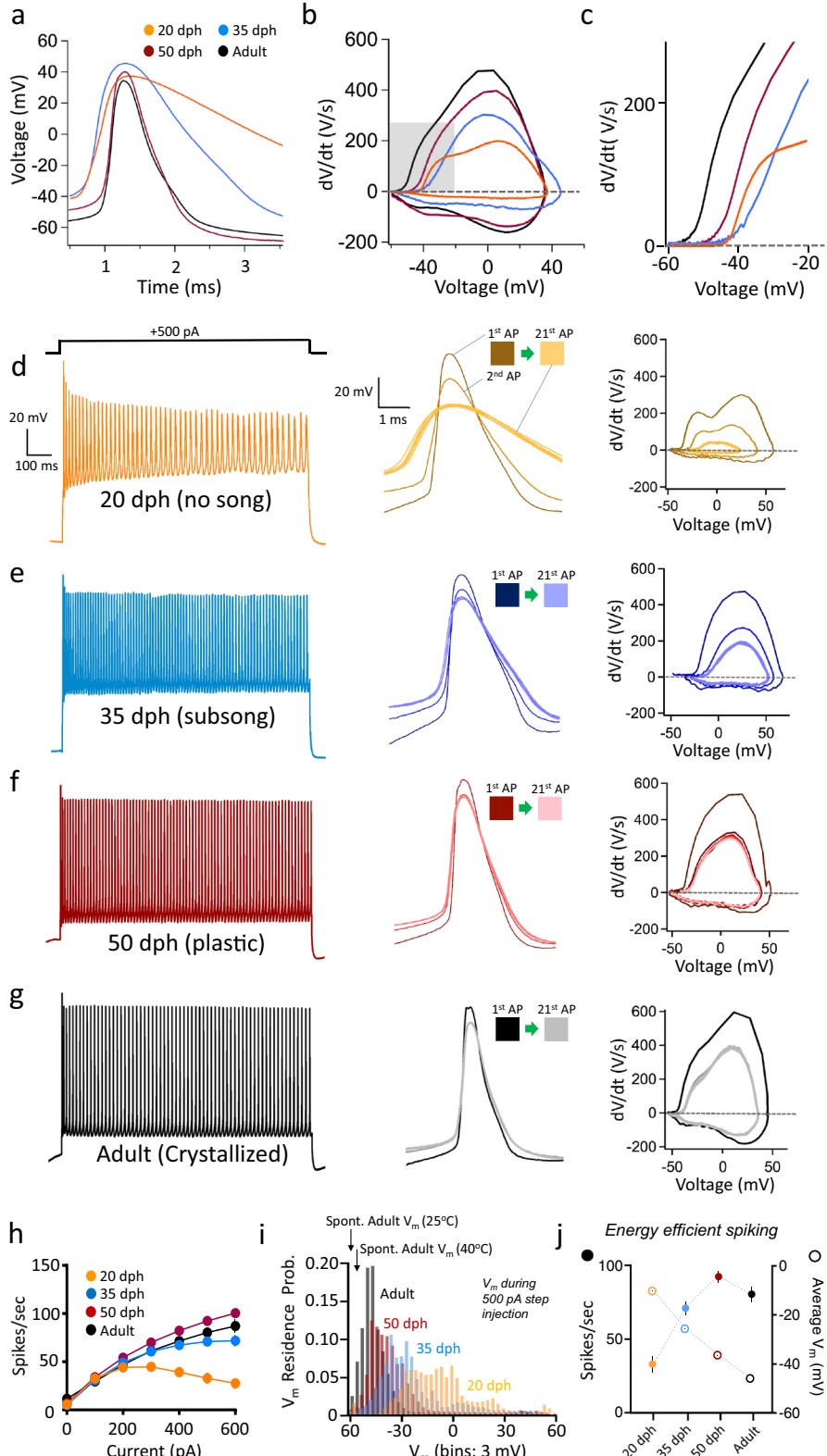

ion channels[78]. We, therefore, performed an additional set of CC experiments in 20 dph and adult finch RAPNs at physiological temperature (~40 °C). Consistent with our room temperature recordings, we found significant differences between 20 dph and adult RA in both spike number (Supplementary Fig. 7a–c) and IFF (Supplementary Fig. 7d) in response to injected current.

**Age-dependent changes in RAPN excitability are not dependent on tutor song exposure.** Male zebra finches raised in the presence of a conspecific male tutor develop songs with high similarity to the tutor song, indicative of vocal imitation. In contrast, males raised under conditions of tutor isolation (i.e., tutor not present) or complete social isolation (i.e., no other birds present) during the vocal learning period develop songs that bear

**Fig. 6 Age-dependent changes in intrinsic excitability of RAPNs in male zebra finches. a** Overlay of representative spontaneous APs recorded across ages (days post hatch, dph) at 24 °C. **b** Overlay of phase plane plots derived from the APs shown in (**a**). **c** Enlarged view of the area highlighted in gray in (**b**). **d**–**g** Left: Representative AP trains elicited by 1 s +500 pA current injection in RAPNs across ages representative of vocal development stages. Center: APs #1, 2, and 16–21 from the traces on the left aligned at the peaks for each age group. Right: Phase plane plots from APs #1, 2, and 16–21 plotted at the same scale for all age groups. **h** Average number of spikes produced during 1 s as a function of current injected for each age group (two-way ANOVA with Tukey's post hoc; $P = 2.7 \times 10^{-8}$, $F$ (18, 494) = 4.189, $N$ (cells/age) = 18/20 dph, 20/35 dph, 20/50 dph, and 14/adults). **i** Probability distribution of membrane potentials ($V_m$) during a 1 s +500 pA current injection across ages. This $V_m$ residence probability is the lowest for adult neurons. Arrows point to the average $V_m$ for spontaneous APs from adult males at ~25 and ~40 °C ($N = 18$ for 20 dph, 20 for 35 dph, 17 for 50 dph, and 14 for adults). **j** Average spikes/sec (Hz; left; full dots) across ages plotted on the same graph as the average $V_m$ (right; open dots). Use legend from (**h**). Data are presented as mean values ± SEM. Spike frequency increases with age, whereas the average $V_m$ decreases.

no resemblance to the songs of the tutored siblings or their tutor[79–81]. Despite the absence of imitation and known changes in song crystallization and critical period closure[66,79,80], lack of exposure to a tutor is not known to affect general measures of vocal behavior (e.g., song rate or the age of onset) or mRNA expression profiles in vocal motor areas (i.e., HVC and RA[80,82]), as long as birds are not prevented from singing[83]. We thus expected that the maturation of RAPN excitable properties we describe here would not be dependent on exposure to tutor song during the critical period for song learning (see, however, Ross et al.[84]). To test this, we compared the electrophysiological properties of 65 dph RAPNs recorded in juvenile males that were normally reared in the presence of their father to those recorded in sibling males that were separated from their fathers at 20 dph, and reared in complete social isolation from 30 to 65 dph. As expected, female-directed songs recorded from tutored birds at 65 dph exhibited a high degree of similarity to the tutor song (mean ± SEM: 72.0 ± 0.5%; Supplementary Fig. 8a, top and middle sonograms), whereas songs from isolated 65 dph birds shared no resemblance to the songs of their father or 65 dph siblings (Supplementary Fig. 8a, bottom sonogram). We also found that isolated and tutored birds showed similarly high degrees of song stereotypy, as reported previously (mean ± SEM: 81.4% ± 0.9 vs. 81.7% ± 1.0; Supplementary Fig. 8b[81]). Patch clamp recordings from the same cohort of sibling birds revealed no differences in the RAPN excitability parameters between isolated and tutored birds, including spontaneous (i.e., spontaneous AP shape and firing frequency; Supplementary Fig. 8c, d), and evoked (i.e., spikes per second and IFF; Supplementary Fig. 8e–g) firing properties. Together, these findings suggest that the development of intrinsic excitable properties in RAPNs is not dependent upon exposure to tutor song and imitation of the tutor song during the critical period of song learning.

**Navβ4 C-terminal peptide induces $I_{NaR}$ and increases the excitability of juvenile RAPNs.** To directly test whether Navβ4 is involved in generating an $I_{NaR}$ in RAPNs, we dialyzed a 14-residue peptide derived from the Navβ4 C-terminus[55] into RA neurons of 20 dph juveniles, shown previously to possess very low $I_{NaR}$ (Fig. 4a–d; Fig. 5d, e). Consistent with a role for Navβ4 in conferring resurgent properties, intracellular dialysis of the β4-WT peptide elicited a large $I_{NaR}$ within several minutes of achieving the whole-cell VC configuration (Fig. 7a, b). This effect was not seen with a scrambled version (β4-Scr, 100 μM) of this peptide (Fig. 7d[55]). The average normalized $I_{NaR}$ to $I_{NaT}$ ratio in RA increased significantly with β4-WT (Fig. 7c) but not with β4-Scr (Fig. 7e) and the fold-change for this ratio differed significantly between cells exposed to β4-WT or β4-Scr (Fig. 7f). We conclude that the channel blocking carboxy-terminal domain of the Navβ4 protein is sufficient to elicit robust $I_{NaR}$ in juvenile RAPNs.

We next examined whether dialyzing 20 dph RAPNs with the β4-WT peptide would make these neurons more excitable.

Indeed, upon dialysis of the β4-WT in whole-cell CC, we detected a prominent "spikelet" that developed in a time-dependent manner during the repolarization phase of the first AP elicited by a positive current injection (Fig. 7g, h). This "spikelet" appeared in 58% of cells (7/12 cells) dialyzed with β4-WT, but never in cells dialyzed with the β4-Scr (0/15 cells) (Fig. 7g, j). Exposure to the β4-WT broadened the AP half-width by 20% (Fig. 7i) and decreased the maximum rate of repolarization by an average of 17% (Supplementary Fig. 9a). Importantly, these effects were not seen with the β4-Scr (Fig. 7k, l; Supplementary Fig. 9b). If we treated "spikelets" as APs, the IFF recorded just after the start of dialysis was not different between groups, but after 15 min the IFF was significantly higher in cells that received the β4-WT than in cells that received β4-Scr (143 ± 21.2 Hz vs. 84.6 ± 3.8 Hz; mean ± SEM; Supplementary Fig. 9c, d).

**In silico modulation of $I_{NaR}$ alters the intrinsic excitability of RAPNs.** As a further test for the role of $I_{NaR}$ in the regulation of excitability in RA, we modeled the $I_{NaR}$ from male RAPNs using a previously published model available in the *Senselab* database as a template[85]. Activation of this modeled $I_{NaR}$ is contingent on brief periods of depolarization followed by hyperpolarization of the membrane potential (see Methods). The experimental data (Fig. 8a) matched the modeled $I_{NaR}$ (Fig. 8b) with a close overlap of $I$–$V$ curves (Fig. 8c) and similar decay kinetics at hyperpolarized potentials (<−20 mV; Fig. 8d).

We predicted that adding this in silico $I_{NaR}$ via dynamic clamp would enhance aspects of excitability in juvenile RA while subtracting it would depress excitability in adult RA. We tested these predictions by comparing the effects of delivering sequential +300 pA current injections at baseline and during dynamic clamping of RAPNs. In 20 dph juveniles, we observed that addition of the modeled $I_{NaR}$ during current injections (Fig. 8e, blue) significantly depolarized the interspike periods, while reducing the interval between the first two spikes (Fig. 8e, red; Table 2) when compared to control (Fig. 8e, black; Table 2). This interspike depolarization was expected given the high input resistance of 20 dph RAPNs (422 MΩ; Table 1). As a result, the IFF increased significantly by 15–20% at all conductances tested (Fig. 8f). These results were reminiscent of our findings using the β4-WT peptide (Supplementary Fig. 9d). This supports a role for $I_{NaR}$ in modulating inter-spike intervals[86]. As with the β4-WT peptide (Fig. 7i and Supplementary Fig. 9a), we also noted an increase in the average AP half-width and a decrease in the average maximum repolarization rate (Table 2). Interestingly, the number of elicited APs decreased in a conductance-dependent manner with the injected in silico $I_{NaR}$ (Table 2), perhaps due to a low density of voltage-gated K$^+$ channels (Kvs) in juveniles that did not allow for a faster recovery of Navs from inactivation.

By contrast, when subtracting the modeled $I_{NaR}$ in adults, we observed hyperpolarization of the membrane potential during the inter-spike periods (Table 3), coinciding with significant decreases in the IFF (5–10%) of the first two spikes when the

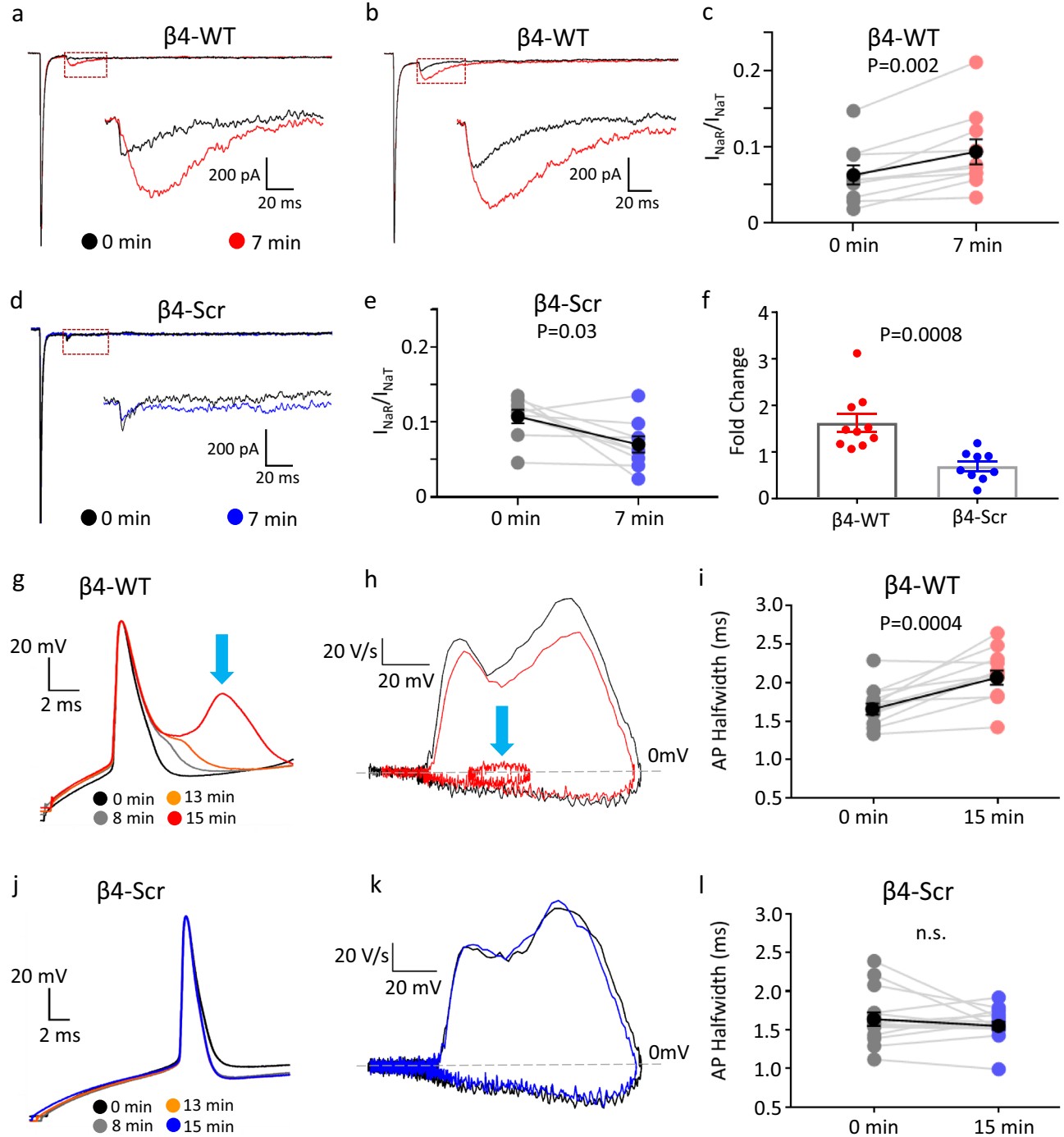

subtracted in silico $I_{NaR}$ conductance was ≥1 mS (Fig. 8g, h). Notably, this modulation did not significantly alter the total number of spikes produced during a 1 s sweep (Table 3). Overall, these findings point to an important role for $I_{NaR}$ in promoting the high-frequency firing of spikes in adult RAPNs.

**Navβ4 mRNA, $I_{NaR}$, and the excitability of RAPNs are sex dependent.** Female zebra finches do not sing. Their RA differs from that of males in developmental trajectory (smaller overall size at 50 dph; Supplementary Fig. 5) and in morphological parameters like cell number and size of soma and processes[61,62]. However, age-dependent changes in the intrinsic excitability of female RAPNs have not been previously examined. In situ hybridization for Navβ4 mRNA showed weak expression and no

evidence of differential labeling in female RA compared to the surrounding arcopallium at 20, 35, or 50 dph (Fig. 9a; refer to Fig. 3 for males). We note that female RA could be unequivocally identified with Nissl staining at the ages tested (Fig. 9b). Contrasting sharply with the marked age-dependent increases of males, we observed a small trend of increased Navβ4 expression after 20 dph (Fig. 9a, c) and a modest, significant increase in the proportion of Navβ4-expressing cells between 20 dph and older ages (Fig. 9a, d). Interestingly, these changes appeared to plateau at 35 dph (Fig. 9c, d). Based on this expression pattern, we predicted that the magnitude of $I_{NaR}$ in female RA would be small across ages. Indeed, we detected a small $I_{NaR}$ at 20 dph and a modest increase at 35 dph, and no further increase at 50 dph (Fig. 9e–g). However, these trends were not statistically significant.

**Fig. 7 Effects of Navβ4 C-terminal peptide (β4-WT) on $I_{NaR}$ and excitability of RAPNs in 20 dph males. a, b** Representative traces of $I_{NaT}$ elicited at +30 mV followed by $I_{NaR}$ elicited at −30 mV immediately after (black) and 7 min after (red) achieving whole-cell configuration with β4-WT (KKLITFILKKTREK) in the patch pipette. The regions in the red dashed boxes are enlarged in the insets. **c** Time-dependent increase in the normalized $I_{NaR}$ to $I_{NaT}$ ratio caused by β4-WT, measured at the −30 mV test potential in individual RAPNs (mean ± SEM in black); two-tailed paired $t$ test, $N = 10$ RAPNs. **d** Representative traces of $I_{NaT}$ elicited at +30 mV followed by $I_{NaR}$ elicited at −30 mV immediately after (black) and 7 min after (blue) achieving whole-cell configuration with β4-Scr (KIKIRFKTKTLELK) in the patch pipette. The region in the red dashed box is enlarged in the inset. **e** Lack of time-dependent increase in normalized $I_{NaR}$ to $I_{NaT}$ ratio by β4-Scr, measured at the −30 mV test potential in individual RAPNs (mean ± SEM in black); paired $t$ test, $N = 9$ RAPNs. **f** Fold changes in normalized $I_{NaR}$ to $I_{NaT}$ ratio between 0 and 7 min for RAPNs exposed to β4-WT or β4-Scr in the patch pipette; two-tailed Student's $t$ test; $N = 10$ and 9 RAPNs for β4-WT and β4-Scr, respectively. Data are presented as individual data points with bars as mean values ± SEM. **g** Representative traces of the first APs elicited by a +300 pA current injection at four-time points after entering the whole-cell configuration with β4-WT in the patch pipette; the APs are aligned at their peak. **h** Phase plane plots from the first elicited AP recorded immediately (black) or 15 min (red) after entering the whole-cell configuration in cells exposed to β4-WT. The blue arrows in **g** and **h** point respectively to the spikelet and resulting alteration in the repolarization rate in the phase plane plot. **i** Time-dependent effects of β4-WT on the 1st AP half-width in individual cells (mean ± SEM in black; two-tailed paired $t$ test; $N = 12$ RAPNs). **j** Representative traces of the first APs elicited by a +300 pA current injection at four-time points after entering the whole-cell configuration with β4-Scr in the patch pipette; the APs are aligned at their peak. **k** Phase plane plots from the first elicited AP recorded immediately (black) or 15 min (red) after entering the whole-cell configuration for cells perfused with the β4-Scr. **l** Time-dependent effects of β4-Scr on the 1st AP half-width in individual cells (mean ± SEM in black; two-tailed paired $t$ test; N = 15 RAPNs).

Whole-cell CC of female RAPNs revealed spontaneous AP firing across all ages, but in contrast to males, there was little evidence of age-dependent changes in active or passive properties (Table 4). These neurons showed increased firing in response to current injections across ages (Fig. 10a–c, left, red traces), but unlike males (Fig. 6), we observed a progressive broadening and amplitude decrease when comparing the 1st, 2nd, and 16th–21st AP at all ages examined (Fig. 10a–c, right). We also failed to detect significant age group differences in elicited spikes per second, in the IFF, or in the fold-change in the maximum depolarization rate between the 2nd and 1st AP (Fig. 10g–i; Supplementary Fig. 10a–c). Phase plots revealed marked differences in rates of depolarization and repolarization between male and female RAPNs at different ages. Specifically, while males had a progressive age-dependent increase in the rate of depolarization during spike trains, the shape of female phase plane plots remained largely unchanged throughout development (Fig. 10d–f). These marked sex differences could be clearly visualized on 3D graphs plotting the relationships between spikes per second, IFF, and the change in the maximum depolarization rate (Fig. 10g–i). Specifically, while 20 dph males and females were tightly clustered together (Fig. 10g), a moderate separation appeared at 35 dph (Fig. 10h), with the sexes occupying distinct clusters by 50 dph (Fig. 10i). Finally, we note the persistence of the double-hump shape of the female AP phase plot across ages (Fig. 10d–f), similar to that of 20 dph males (Fig. 6b; Fig. 10d, right), and in contrast to 35 and 50 dph males. The lack of change in this waveform suggests a persistently low density of Nav channels in the female RAPNs at the axon initial segment and in the soma-dendritic compartment[43].

Principal component analysis (PCA) applied to multiple spontaneous AP parameters from male and female RA neurons across ages provided quantitative support and further insights into these marked developmental sex differences. PCA1, which accounted for 53% of the total variance, segregated males but not females according to age, with 50 dph males localizing to the right quadrants, whereas younger males and all females from all age groups localized mostly to the left quadrants (Supplementary Fig. 11, $x$-axis). Interestingly, PCA1 accounted for >80% of the variance of the AP half-width and the maximum rates of depolarization and repolarization, with the half-width showing a strong negative correlation with the other two parameters. PCA2 and PCA3 accounted respectively for 20%, and 17% of the total variance. PCA2 predominantly accounted for the variance of the AP amplitude and threshold, which showed a strong negative correlation, and PCA3 almost exclusively accounted for the

variance in the spontaneous firing frequency, but neither PCA2 (Supplementary Fig. 11, $y$-axis) nor PCA3 helped segregate the data by age groups.

## Discussion

Song production in the zebra finch requires fast and precise signaling from RAPNs to activate and suppress syringeal musculature[31]. Our results reveal that Navβ4 underpins a robust $I_{NaR}$ that facilitates precise, prolonged, and reliable high-frequency firing. We propose that this mechanism may allow RAPNs to operate as highly specialized motor control units via burst-and-pause spike firing[31]. Given the prominent expression of Navβ4 in mammalian deep cortical layers[87,88], we suggest that $I_{NaR}$ may be more prevalent in neocortical L5PNs than previously suspected. Our findings thus provide insights into the unique molecular building blocks of specialized upper motor neurons that drive fine motor control.

**RAPNs possess unusually large and fast Na$^+$ currents**. We have shown that AP waveforms in mature RAPNs have: (1) low spike thresholds, (2) ultra-narrow half-widths, (3) non-adapting amplitudes at high firing frequencies, and (4) large maximum depolarization rates (Fig. 1 and Table 1). Low spike thresholds of RAPNs may ensure an effective response to the sparse coding of HVC inputs[89]. Low thresholds require robust Na$^+$ currents[90]. Indeed, our data revealed a remarkably large peak $I_{NaT}$ (max. $I_{Na} = 61$ nA at ~40 °C; Table 1) during the AP upstroke. By comparison, fast-spiking neurons of similar size in deep cerebellar nuclei have only 21 nA[73]. RAPNs also possess unusually large peak $I_{NaR}$ (~3.0 nA at ~24 °C) compared to the calyx of Held nerve terminals, MNTB principal neurons, and Purkinje neurons in mouse, and brainstem neurons in chicken, which exhibit peak $I_{NaR} \leq 0.2$–0.5 nA[55,58,71,86]. Together with a fast $I_{NaT}$, this large $I_{NaR}$ facilitates the high-frequency firing of spikes with ultra-narrow half-widths in RAPNs (Fig. 1g).

**RAPNs operate like specialized pyramidal neurons in motor cortex**. Our study reveals several striking similarities between the intrinsic excitability of RAPNs and "regular spiking" L5PNs found in the mammalian motor cortex[50,91,92]. Like L5PNs[25], RAPNs have an $I_h$-mediated sag[40], a persistent Na$^+$ current, and fast non-linear activation of $I_{NaT}$ with a short onset latency[69] (see Fig. 5g) that all likely contribute to "regular spiking" in both cell types. Like RAPNs, rodent[93–95] and human[96] cortical neurons exhibit rapid-onset spikes with low thresholds, probably caused

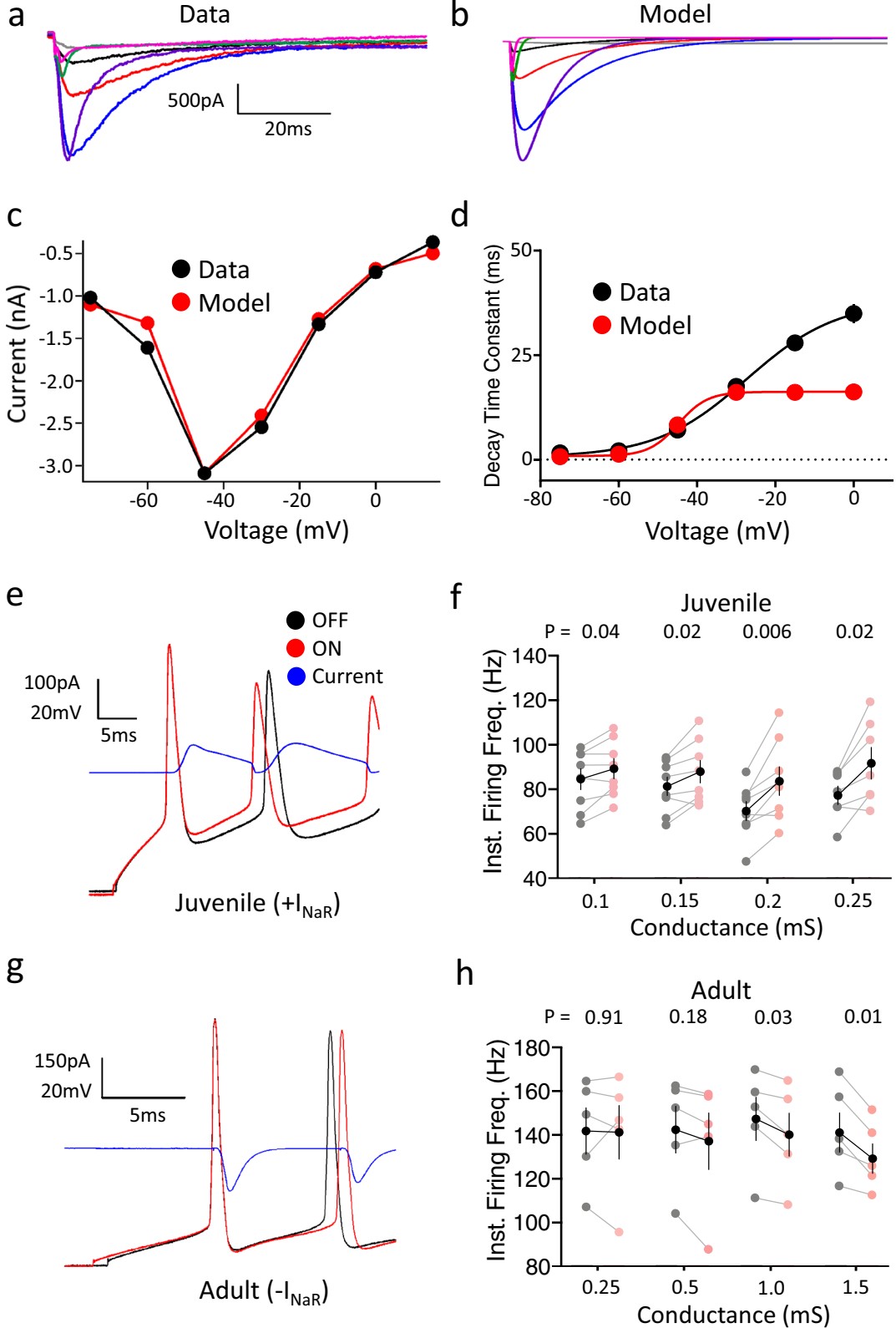

by a high density of Na$^+$ channels at a distal axonal spike initiation site[94,95]. Yet another shared feature is the presence of a biphasic depolarization in the AP phase plane plots that reveals an abrupt "kink" followed by a soma-dendritic component[43] (see Fig. 1e). The developmental profile of RAPNs also closely resembles that of maturing rat L5PNs, including narrowing of spike waveforms and increases in spike firing frequency[26]. Lastly,

similar to RAPNs, cells in L5 of mouse motor cortex express high levels of Navβ4 mRNA[87], suggesting that L5PNs may have $I_{NaR}$ and that these currents may be more prevalent in the neocortex than previously suspected.

Despite these similarities, RAPNs also possess specialized properties that distinguish them from typical L5PNs. In particular, adult RAPNs are capable of producing remarkably

**Fig. 8 Dynamic clamping of spiking RAPNs in juvenile and adult male finches. a** Family of recorded $I_{NaR}$ (same as in Fig. 2c). **b** Family of modeled $I_{NaR}$ currents (same voltage protocol that elicited currents in Fig. 2c; same scale as in (**a**)). **c** Overlay of the current-voltage relationship of peak $I_{NaR}$ comparing the modeled (red) and average recorded values (black; same as in Fig. 2e). **d** Overlay of the time constant of decay ($\tau$) as a function of voltage in the modeled (red) and recorded currents (black). **e** APs elicited in 20 days post hatch (dph) RAPNs before (OFF; black) or during dynamic clamp-mediated addition (ON; red) of the in silico $I_{NaR}$ (blue) peaking during the AP repolarization phase. APs were triggered by a +300 pA current injection. **f** Paired data of the instantaneous firing frequency (Inst. Firing Freq.) change recorded in 20 dph RAPNs upon dynamic clamp-mediated addition (ON; pink) compared to the control (OFF; gray) condition as a function of conductance (mean ± SEM in black, two-tailed paired $t$ test, $P$-values in graph; $N = 8$ RAPNs). **g** APs elicited in adult RAPNs before (OFF; black) or during dynamic clamp-mediated subtraction (ON; red) of the in silico $I_{NaR}$ (blue) peaking during the AP repolarization phase. APs were triggered as in (**e**). **h** Paired data of the instantaneous firing frequency change recorded in adult RAPNs upon dynamic clamp-mediated subtraction (ON; pink) compared to the control (OFF; gray) condition as a function of conductance (mean ± SEM in black, two-tailed paired $t$ test, $P$-values in the graph; $N = 5$ RAPNs).

narrow APs (half-width = 0.18 ms at 40 °C; Fig. 1g), which makes them well suited to integrate and encode bursts of high-frequency synaptic inputs at their multiple spiny dendrites[24]. In contrast, the APs of L5PNs found in the M1 of cat[11] (half-width: 0.42 ms), mouse[2] (half-width: 0.65 ms), and rat[91] (half-width: ~0.8 ms) are markedly broader. These differences in the spiking properties of L5PNs across species may be due in part to differences in the amplitudes of $I_{NaT}$ and Kv currents, as well as the size of $I_{NaR}$.

Intriguingly, the fast-spiking of RAPNs is perhaps rivaled only by the large Betz-type L5PNs found in the M1 cortex of awake macaque that exhibit narrow AP spikes (half-width: 0.26 ms)[9] with fast conduction velocities (~80 m/s) along large-caliber axons[10]. Betz-type cells are corticospinal neurons that are likely important for voluntary fine motor control[10]. While it is not known whether these cells express Navβ4 or $I_{NaR}$, there is evidence that they express high threshold Kv3.1 channels[10], which are associated with fast-spiking neurons with narrow APs[97,98]. Interestingly, Kv3.1 is notably absent in rodent L5PNs[10] and *KCNC1*, the gene that encodes Kv3.1, is a strong molecular marker of RA[99,100]. Recently, Navβ4 has been localized to a sparse subpopulation of human L5PNs that project to extra-telencephalic areas[101]. Taken together, these results suggest a remarkable evolutionary convergence of molecular and physiological properties in a subclass of extratelencephalic projecting neurons that feature prominently in songbirds and primates, whose properties likely evolved to enable precise spike timing for fine motor control[31].

**Fast $I_{NaT}$ inactivation may promote energy-efficient spikes.** RAPNs show a profound decrease in the time constant of $I_{NaT}$ inactivation during development that correlates with both narrowing of the AP and increases in spike frequency (Figs. 5h and 6). In mammals, rapid inactivation of Nav channels can narrow the AP duration by limiting the overlap of $Na^+$ and $K^+$ conductances[74,75]. While this narrowing can increase the temporal precision of spiking, it also serves an important role in mitigating the high metabolic demand of fast-spiking neurons[34,49]. By quickly returning the membrane potential ($V_m$) during the AP to more hyperpolarized potentials, fast-spiking neurons can limit the residency time of Nav channels in open states, thus lessening the burden of the $Na^+/K^+$-ATPase in restoring ionic gradients[76]. Importantly, in RAPNs, we found that despite a 3× increase in evoked spike frequency during development, the distribution of the $V_m$ shifted to hyperpolarized voltages at older ages (Fig. 6i). This strongly suggests that like mammals[76], the zebra finch utilizes fast $I_{NaT}$ inactivation, likely facilitated by Navβ4, to increase the energy efficiency of fast-spiking neurons. In addition, while previous studies have described AP narrowing in L5PNs of rodents[26], this analysis provides a novel way of evaluating the dynamic changes of $V_m$ in spiking neurons during development.

**Developmental changes in RAPN intrinsic properties promote reliable AP bursts.** Previous studies have attributed major in vivo changes in RA activity during the vocal learning period primarily to changes in the synaptic strength of inputs from HVC and LMAN[32,41,48,102]. However, we observed significant changes between 35 and 50 dph juveniles in several intrinsic excitability features of RAPNs during high-frequency firing (Fig. 6 and Table 1). Furthermore, while the spike frequency during positive current injections did not change between 50 dph juveniles and adults[41], we found significant changes in spike threshold and waveform between these ages (Table 1; Fig. 6; Supplementary Fig. 6). The complete developmental switch of Navβ3 for Navβ4 in RA strongly correlated with the appearance of a larger $I_{NaT}$ and $I_{NaR}$, leftward shifts in $I_{NaT}$ activation, and faster $I_{NaT}$ inactivation, all properties thought to facilitate high-frequency spike firing. We, therefore, propose that in addition to synaptic changes, significant changes to intrinsic excitability of RAPNs occur and may contribute to the developmental increase in precise burst-and-pause spike firing observed in vivo in RAPNs during song production[31,32,44,103]. Notably, we refer in this study to changes in excitable and molecular properties of RAPNs as age-dependent, but further studies are needed to determine whether these changes reflect a developmental program intrinsic to RA or whether they are affected by other factors that change with age, like the extent of song practice and singing experience.

Importantly, the development of RAPN excitable properties was not significantly affected by access to a tutor or social contact (Supplementary Fig. 8), suggesting that these properties are not dependent on exposure to tutor song or vocal imitation. We note that isolation from a tutor is not thought to affect general measures of vocal behavior like singing rates or age of onset, and the isolate songs still contain defined syllables with relatively complex acoustic structure and stereotypy, with vocal-motor development in these birds likely guided by auditory feedback from their own vocalizations[80,81,83]. Our results thus show that RAPN excitable properties at 65 dph are unaffected by whether the descending motor commands received by RA result from vocal imitation vs. vocal practice of self-referential (isolate) song. Future studies will be needed to fully determine whether developmental changes in the excitability of RAPNs reflect a cell-autonomous program or depend on extrinsic factors like intact hearing and/or synaptic inputs to RA.

**Auxiliary Navβ subunits help shape spike waveforms.** Adult male finches show higher expression of Nav1.6 (*SCN8A*) in RA compared to the surrounding arcopallium[52]. Nav1.6 carries the majority of $I_{NaR}$ in a number of cell types[56]. We suggest that a combination of fast inactivation mediated by this α subunit (a hinged-lid mechanism), together with the blocking domain of the

**Table 2 Average AP properties from juvenile male dynamic clamp experiments.**

| | 0.1 mS | | 0.15 mS | | 0.2 mS | | 0.25 mS | |
|---|---|---|---|---|---|---|---|---|
| | OFF | ON | OFF | ON | OFF | ON | OFF | ON |
| # APs | 44.9 ± 3.9 | 43.6 ± 3.9* (P = 0.02) | 44 ± 4.0 | 39.1 ± 5.1* (P = 0.01) | 37.4 ± 3.7 | 26.9 ± 4.6* (P = 0.006) | 44.2 ± 3.4 | 35 ± 4.37* (P = 0.01) |
| Amp. (mV) | 57.2 ± 3.0 | 44.1 ± 7.3 | 62 ± 2.8 | 52.9 ± 2.8* (P = 0.0005) | 67.5 ± 2.4 | 55.7 ± 2.8* (P = 0.002) | 65.7 ± 1.8 | 50.4 ± 2.8* (P = 0.0003) |
| HW (ms) | 3.2 ± 0.2 | 3.0 ± 0.4 | 2.9 ± 0.7 | 3.1 ± 0.2* (P = 0.03) | 2.5 ± 0.1 | 2.7 ± 0.2 | 2.6 ± 0.1 | 3.0 ± 0.3 |
| Max D. (V/s) | 84.2 ± 9.1 | 67 ± 8.1* (P = 0.004) | 96.4 ± 9.4 | 72.7 ± 7.0* (P = 0.001) | 118.2 ± 9.6 | 85.6 ± 8.2* (P = 0.0007) | 109.6 ± 6.6 | 71.9 ± 6.5* (P = 0.0002) |
| Max R. (V/s) | 28.7 ± 2.9 | 25.1 ± 2.2* (P = 0.05) | 30 ± 2.2 | 26.9 ± 2.1* (P = 0.002) | 32.2 ± 2.3 | 30.1 ± 2.4 | 32 ± 1.4 | 27.6 ± 1.9* (P = 0.006) |
| Peak (mV) | 22.5 ± 3.4 | 18.2 ± 3.2* (P = 0.002) | 24.9 ± 3.2 | 18.8 ± 3.3* (P = 0.00004) | 28.1 ± 2.6 | 18.7 ± 3.1* (P = 0.00004) | 32.7 ± 2.4 | 25.5 ± 2.9* (P = 0.002) |
| AHP (mV) | −39.7 ± 1.8 | −35.2 ± 3.1* (P = 0.02) | −40.3 ± 2.5 | −36.8 ± 3.1* (P = 0.006) | −42.6 ± 1.5 | −37.4 ± 2.0* (P = 0.0002) | −44.2 ± 1.6 | −38.6 ± 2.2* (P = 0.004) |

Values at the top of each paired data set are conductance values used during dynamic clamping. Values in the OFF column represent values obtained during a +300 pA current injection without dynamic clamping. Values in the ON column represent values obtained during the same current injection with dynamic clamping. For comparisons, a two-tailed paired t test was used for each conductance comparing values obtained before and during dynamic clamping. # APs—the number of APs produced during a 1 s +300 PA current injection, Amp.—AP amplitude, HW—AP half-width, Max. D.—maximum depolarization rate, Max. R.—maximum repolarization rate, Peak—AP peak, AHP—AP afterhyperpolarization peak. N = 7 cells.

covalently bound Navβ4 may greatly accelerate inactivation kinetics (Fig. 5h). In addition, the Navβ4 blocking peptide is thought to protect Nav channels from entering a long-lived inactivation state[56]. A large surplus of Nav channels available for opening is critical for limiting cumulative inactivation, thus promoting reliable, high-frequency AP firing with little amplitude attenuation[104]. Interestingly, coexpression of Nav1.6 and Navβ4 in HEK cells causes a hyperpolarizing shift in activation (~8 mV)[105] similar to what we observed in Fig. 5f (a ~10 mV hyperpolarizing shift of the Na$^+$ current IV curve from 20 dph to adult). Without a high density of available Nav channels, RAPNs would exhibit smaller spike amplitudes, prolonged interspike intervals, and AP failures during high-frequency firing[104] as seen in 20 dph juveniles (Fig. 6) and in arcopallial neurons outside RA in adults (Fig. 2).

Adult RA also shows high expression of Navβ1, which binds the Nav α subunit non-covalently[106]. In mammals, Navβ1 promotes the localization of Nav1.6 to the axon initial segment of L5PNs[107] and contributes to $I_{NaR}$ in cerebellar granule cells[108]. Nav1.6 exhibits fast inactivation[109,110] and co-expression with Navβ1 greatly accelerates recovery from inactivation, as well as expression of a small persistent current (~5% of peak current at +20 mV)[110]. We note that finch Navβ1 shares ~70% conserved residues with rodents, and both Navβ1 and 4 are recognized as members of the Navβ family of modulatory subunits by predictive algorithms (e.g., InterPro scan), indicative of a conserved molecular structure with mammals. It thus seems likely that high expression of Nav1.6, combined with Navβ1 and Navβ4 may contribute to generating large Na$^+$ currents with fast activation and inactivation kinetics in RAPNs[105,109]. Of note, Navβ2 has not been linked to $I_{NaR}$ and is not differentially expressed across the arcopallium[52].

**$I_{NaR}$ facilitates prolonged and precise high-frequency firing in RAPNs.** Prominent $I_{NaR}$ and fast $I_{NaT}$ inactivation kinetics are also observed in the mature calyx of Held nerve terminals and MNTB principal neurons[71,86,111], which exhibit narrow AP spikes and can fire at high frequencies (up to 1 kHz)[112]. Moreover, $I_{NaR}$ generates a depolarizing after-potential at the calyx of Held once the spike downstroke has terminated[71], similar to that seen in RAPN spikes (Fig. 1c). Importantly, this depolarizing after-potential that follows the AP spike sets the periodicity of the spontaneous spiking (Fig. 1b). Dynamic clamp experiments revealed that the addition of an in silico $I_{NaR}$ increases the instantaneous firing frequency (IFF) of 20 dph RAPNs. Conversely, subtraction of $I_{NaR}$ significantly decreased the IFF of adult RAPNs (Fig. 8). Because the addition or subtraction of a modeled $I_{NaR}$ has no direct effect on Nav channel availability, like a peptide blocker would, these results provide compelling evidence that $I_{NaR}$ alone is sufficient to enhance the firing frequency of RAPNs.

Although $I_{NaR}$ has been linked to Navβ4 in a number of cell-types[55,71,113,114], such a link has been disputed in recent studies where knock-out of the Navβ4 gene failed to eliminate this current in cerebellar Purkinje neurons[115,116]. However, the knockout was constitutive and alternative compensatory mechanisms may have enabled this $I_{NaR}$ current[116]. Among recently identified alternatives, FGF14 may provide primary and/or redundant roles for promoting $I_{NaR}$ in different cell types[116,117]. However, FGF14 is not differentially expressed or higher in RA compared to other arcopallium domains[118]. Thus, in sharp contrast with Navβ4, FGF14 expression does not correlate with the spatial distribution of $I_{NaR}$ in finches. In our study, the β4-WT peptide was sufficient to increase $I_{NaR}$ and the IFF of RAPNs from pre-song juvenile finches, providing compelling evidence of a

**Table 3 Average AP properties from adult male dynamic clamp experiments.**

| | 0.25 mS | | 0.5 mS | | 1.0 mS | | 1.5 mS | |
| --- | --- | --- | --- | --- | --- | --- | --- | --- |
| | OFF | ON | OFF | ON | OFF | ON | OFF | ON |
| # APs | 69.4 ± 8.7 | 49.2 ± 8.1 | 73.6 ± 7.6 | 72.6 ± 11.4 | 79 ± 8.2 | 76.6 ± 8.9 | 72 ± 10.0 | 67.6 ± 7.9 |
| Amp. (mV) | 93.1 ± 2.4 | 99.3 ± 2.9 | 94.6 ± 29.2 | 94.1 ± 2.7 | 91.2 ± 1.7 | 94.3 ± 2.6 | 94.2 ± 1.9 | 95.5 ± 2.2 |
| HW (ms) | 0.57 ± 0.02 | 0.58 ± 0.03 | 0.57 ± 0.27 | 0.57 ± 0.03 | 0.5 ± 0.03 | 0.6 ± 0.03 | 0.6 ± 0.03 | 0.6 ± 0.03 |
| Max D. (V/s) | 487.54 ± 19.3 | 390.6 ± 32.3 | 519.3 ± 108.5 | 531.3 ± 27.5 | 535.6 ± 24.5 | 548.3 ± 25.7* ($P = 0.03$) | 545.8 ± 24.8 | 542.2 ± 38 |
| Max R. (V/s) | 191.4 ± 11.4 | 163.4 ± 30.0 | 192.7 ± 54.2 | 195.3 ± 16.4 | 197.1 ± 14.9 | 199.9 ± 15.4 | 199.9 ± 15.9 | 201.8 ± 16.8 |
| Peak (mV) | 38.08 ± 1.1 | 38.3 ± 1.4* ($P = 0.008$) | 40.5 ± 21.4 | 41.6 ± 2.3 | 40.3 ± 1.2 | 40.6 ± 1.3* ($P = 0.006$) | 40.96 ± 1.2 | 20.4 ± 0.9 |
| AHP (mV) | −62.7 ± 1.0 | −65.5 ± 1.4 | −63.4 ± 1.0 | −65.1 ± 0.93* ($P = 0.03$) | −64 ± 1.0 | −65.8 ± 0.9* ($P = 0.002$) | −64.4 ± 0.9 | −67.5 ± 0.7* ($P = 0.004$) |

Values at the top of each paired data set are conductance values used during dynamic clamping. Values in the OFF column represent values obtained during a +300 pA current injection without dynamic clamping. Values in the ON column represent values obtained during the same current injection with dynamic clamping. For comparisons, a two-tailed paired *t* test was used for each conductance comparing values obtained before and during dynamic clamping. # APs—the number of APs produced during a 1 s +300 PA current injection, Amp.—AP amplitude, HW—AP half-width, Max. D.—maximum depolarization rate, Max. R.—maximum repolarization rate, Peak—AP peak, AHP—AP afterhyperpolarization peak. $N = 5$ cells.

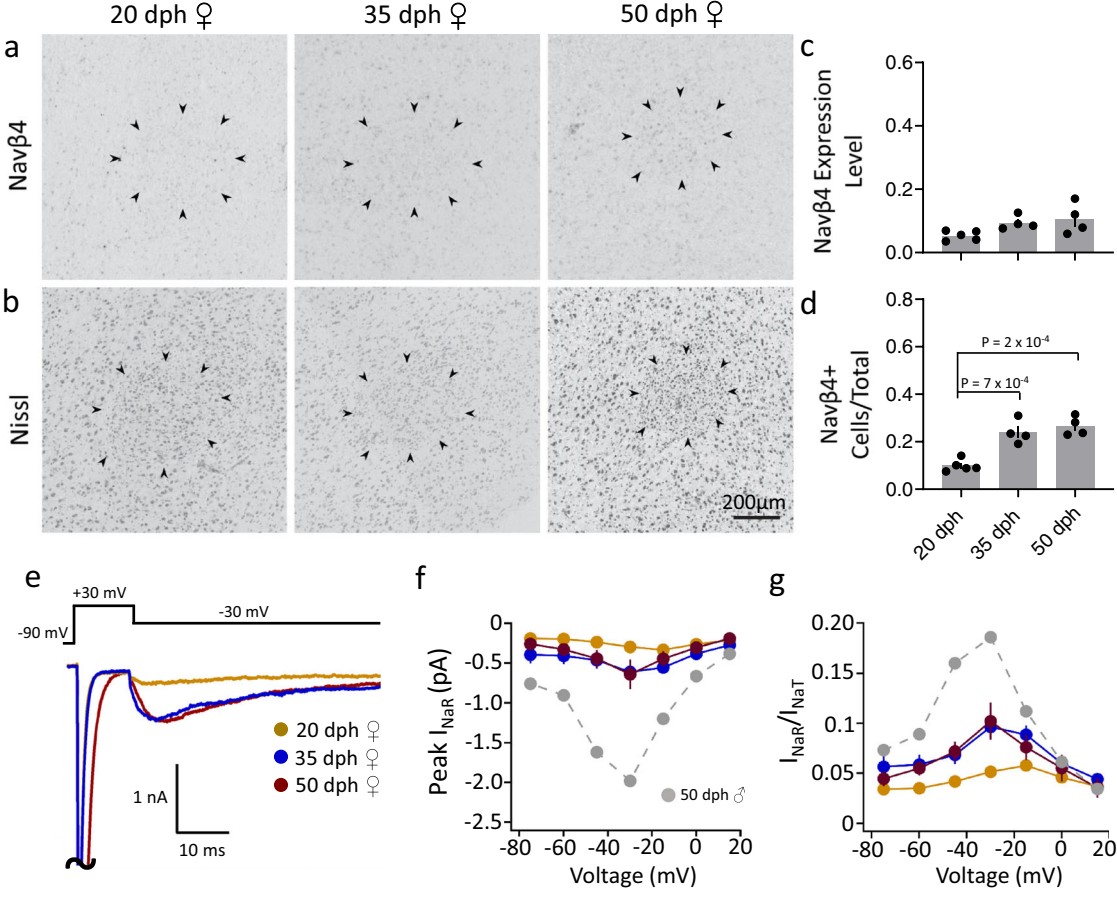

**Fig. 9 Navβ4 mRNA and intrinsic excitable properties in RA of female zebra finches across ages. a** Representative in situ hybridization images for Navβ4 across ages (days post hatch, dph); black arrowheads indicate RA borders. **b** Views of adjacent sections to those in (**a**), stained for Nissl; black arrowheads indicate RA borders. **c** Comparison of Navβ4 expression levels (normalized optical density) across age groups within RA (one-way ANOVA with Tukey's post hoc; $P = 0.06$, $F (2, 10) = 3.290$, $N$ (birds/age) = 5/20 dph, 4/35 dph, and 4/50 dph females). Data are presented as individual data points with bars as mean values ± SEM. **d** Comparison of the proportions of Navβ4-expressing cells across age groups within RA (one-way ANOVA with Tukey's post hoc; $P = 0.0001$, $F (2, 10) = 24.95$; $N$ (birds/age) = 5/20 dph, 4/35 dph, and 4/50 dph females). Data are presented as individual data points with bars as mean values ± SEM. **e** Representative currents from RAPNs at each age group during the −30 mV test potential shown at the top. The large $I_{NaT}$ peaks have been truncated. **f** Average I–V curves for the peak $I_{NaR}$ in RA neurons at each age group; no significant groups differences seen (two-way ANOVA with Tukey's post hoc; $P = 0.83$, $F (12, 140) = 0.6103$, $N$ (cells/age) = 9/20 dph, 10/35 dph, and 4/50 dph females). Data are presented as mean values ± SEM. Gray dashed line shows I–V relationship for 50 dph male (same as in Fig. 4c). **g** I–V curves after normalization of the peak $I_{NaR}$ to the peak $I_{NaT}$ measured in a given sweep then averaged across cells (two-way ANOVA with Tukey's post hoc; $P = 0.48$, $F (12, 140) = 0.9678$, $N$ (cells/age) = 9/20 dph, 10/35 dph, and 4/50 dph females). Data are presented as mean values ± SEM. Gray dashed line shows I–V relationship for 50 dph male (same as in Fig. 4d).

**Table 4 Passive and spontaneously active properties of female arcopallial neurons.**

| Female ♀ | | | |
|---|---|---|---|
| Age (dph) | 20 | 35 | 50 |
| Input Res. (MΩ) | 442.3 ± 34.2[a,b] $N = 19$ | 255.1 ± 26.2[c] $N = 24$ | 262.6 ± 22.5[c] $N = 8$ |
| % Spont. | 52.4 | 43.4 | 42.8 |
| Spont. Freq. (Hz) | 2.5 ± 0.7 $N = 11$ | 4.7 ± 1.5 $N = 10$ | 5.3 ± 2.1 $N = 3$ |
| Thresh. (mV) | −46.5 ± 1.8 $N = 11$ | −44.1 ± 1 $N = 10$ | −47.9 ± 0.8 $N = 3$ |
| HW (ms) | 1.7 ± 7E10−2 $N = 11$ | 1.6 ± 9E10−2 $N = 10$ | 1.5 ± 2E10−1 $N = 3$ |
| Amp. (mV) | 88.2 ± 2.6 $N = 11$ | 78.5 ± 2.9 $N = 10$ | 78.9 ± 1.9 $N = 3$ |
| Max. D. (V/s) | 247.5 ± 12.2 $N = 11$ | 210.1 ± 19.6 $N = 10$ | 255.8 ± 29.9 $N = 3$ |
| Max. R. (V/s) | 49.7 ± 2.5[b] $N = 11$ | 52 ± 4.8[b] $N = 10$ | 63.6 ± 11.5[c,a] $N = 3$ |

A one-way ANOVA with a Tukey post hoc test was used for each measurement across the three ages (days post hatch, dph) indicated. Input Res.—input resistance, $P = 2.2 \times 10^{-5}$, $F_{(2,48)} = 13.5$, % Spont.—proportion of cells that had spontaneous APs; Spont. Freq.—spontaneous firing frequency, $P = 0.29$, $F_{(2, 41)} = 1.262$; Thresh.—AP threshold, $P = 0.61$, $F_{(2, 21)} = 0.5043$; HW—AP half-width, $P = 0.54$, $F_{(2, 21)} = 0.6422$, Amp.—AP amplitude, $P = 0.15$, $F_{(2, 21)} = 2.065$; Max. D.—maximum depolarization rate, $P = 0.42$, $F_{(2, 21)} = 0.9094$; Max. R.—maximum repolarization rate, $P = 8.7 \times 10^{-10}$, $F_{(2, 21)} = 66.05$.
[a]Data is significant different from 35 dph ($P \leq 0.05$).
[b]Data is significant different from 50 dph ($P \leq 0.05$).
[c]Data is significantly different from 20 dph ($P \leq 0.05$).
Input resistance: 20 dph vs. 35 dph ($P = 2.8 \times 10^{-5}$), 20 dph vs. 50 dph ($P = 0.003$).
Maximum repolarization rate: 20 dph vs. 50 dph ($P = 1.1 \times 10^{-9}$), 35 dph vs. 50 dph ($P = 1.7 \times 10^{-9}$).

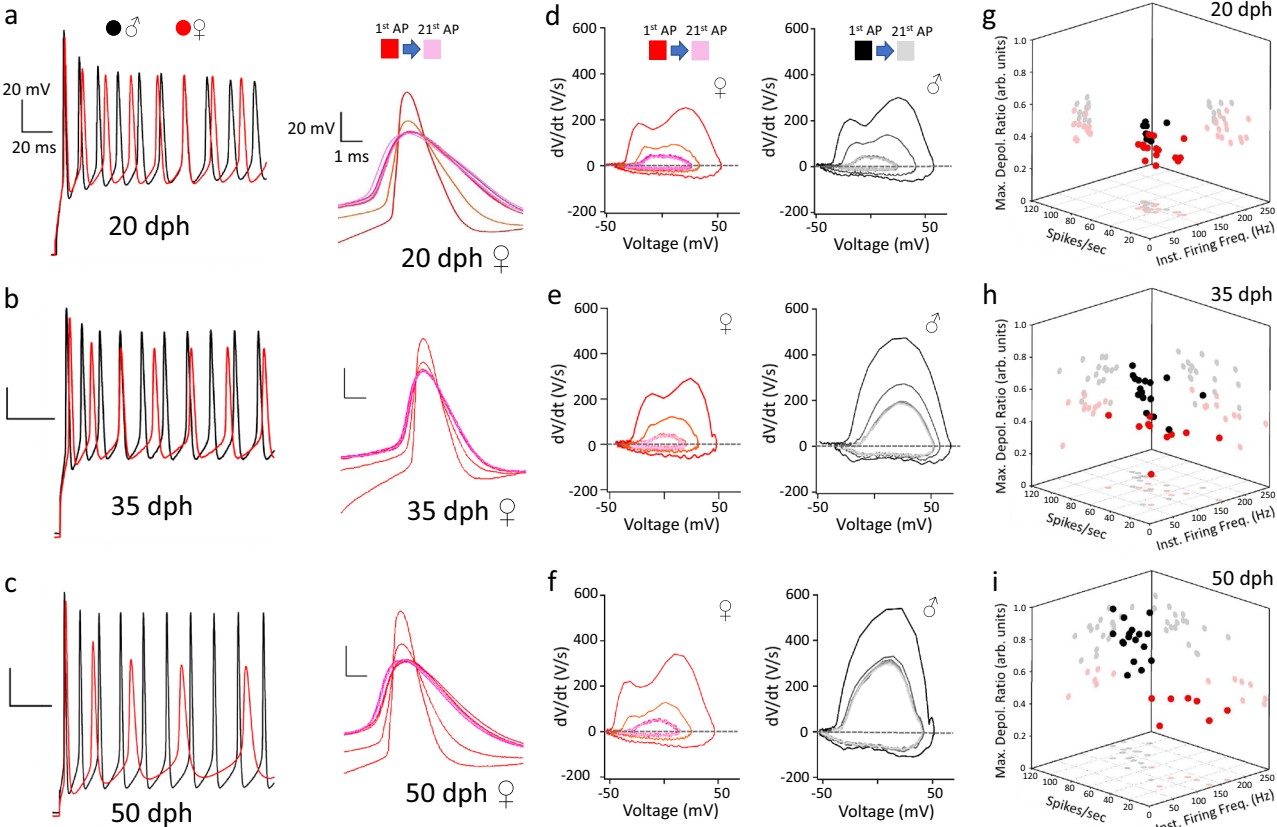

**Fig. 10 Age-dependent changes in intrinsic excitability of RAPNs in female and male zebra finches. a–c** Left: Representative AP trains elicited by a +500 pA current injection in RAPNs in males (First 10 APs; black; same as in Fig. 6d–f, left) and females (red) across ages (days post hatch, dph). Right: APs #1, 2, and 16–21 from the traces on the left aligned at the peaks for each age group in females. **d–f** Phase plane plots from APs #1, 2, and 16–21 plotted at the same scale for all age groups for females (left) and males (right; same as in Fig. 6d–f, right). **g–i** 3D dot plots showing the instantaneous firing frequency (Inst. Firing Freq.), number of spikes per second (Spikes/sec), and the fold-change in maximum depolarization rate (Max. Depol. Ratio; arbitrary units, arb. units) between the 2nd and 1st APs during a 1 s +500 pA current injection in male (black) and female (red) RAPNs. $N$ (cells/age) = 21/20 dph, 23/35 dph, and 7/50 dph females. Refer to Fig. 6 for male data.

causal link between Navβ4 and these excitable properties. Most recordings resulted in a second 'spikelet' before the first AP had fully repolarized, in some cases resembling the after-depolarization seen in adult RAPNs (Fig. 1c). A similar effect is also seen at the rodent calyx of Held when this peptide is dialyzed into immature nerve terminals[71]. Fast AP repolarization rates are

absent in 20 dph finches (Table 1), which may explain why we did not see a time-dependent increase in the number of spikes in these peptide experiments. We suggest that with a faster AP repolarization rate, these "spikelets" would have produced full-blown spikes that would have more closely resembled the mature AP waveform and firing.

**Sex differences in the intrinsic excitability of maturing RAPNs.**
Although largely unexplored, defining sex differences in the excitability of zebra finch song nuclei may help to identify critical requirements for neurons that control learned vocalizations. Interestingly, the developmental changes in Navβ4 expression, $I_{NaR}$, spike waveforms and firing rates of RAPNs seen in males were absent in females, which do not sing (Figs. 9, 10). These observations provide further support that Navβ4 expression underlies $I_{NaR}$. While we acknowledge that a number of variables under the complex regulation of genetic and hormonal factors likely contribute to the lack of female vocal learning[61,119], we suggest that the lack of maturation in the intrinsic properties of RAPNs could also contribute to females being unable to produce complex, learned vocalizations.

**Navβ subunits promote neuronal excitability in RAPNs.** Our results point to a key piece of "hardware" that allows adult male RAPNs to function as precise and reliable generators of fast-spiking. High expression of Navβ4, likely coupled with Navβ1 and Nav1.6α subunits, enables a large $I_{NaR}$ and an $I_{NaT}$ with fast activation and inactivation. These features lower the AP threshold and allow robust, rapid-onset APs with large non-adapting amplitudes[95]. Together with Kv3.1 channels[10,97,120], this combination of Navα/β subunits narrows the spike half-width and enables fail-safe fast-spiking activity to emerge during the critical song learning period. By contrast, juvenile and female[100] RAPNs express high levels of Navβ3, do not have a large $I_{NaR}$, and cannot spike at high frequencies. Finally, we note a possible evolutionary convergence in firing properties between songbird RAPNs and Betz-type L5PNs in primate motor cortex[9,10] that may have fine-tuned them for rapid and precise motor control[3]. We thus propose that the specialized "hardware" of RAPNs enables their precise burst-and-pause firing, which is required for activation of lower motor neurons that control the superfast syrinx muscles involved in singing[33].

## Methods

**Animal subjects.** All of the work described in this study was approved by OHSU's Institutional Animal Care and Use Committee and is in accordance with NIH guidelines. Zebra finches (*Taeniopygia guttata*) were obtained from our own breeding colony or purchased from local breeders. The ages of birds in the 20, 35, and 50 days post hatch (dph) groups were established based on the date the first egg hatched, and thus the birds were sacrificed within ± 2 days of the target age. Birds older than 120 dph were considered adults. On the day of the experiment, all of the birds with the exception of 20 dph birds, were removed from the colony at lights-on (10 AM PST) and housed for ~1 h in an acoustic isolation chamber to minimize the potential confounds of singing and non-specific auditory stimulation on gene expression, noting that social isolation can also alter neurogenomic state[121]. The 20 dph finches were removed directly from the colony, and not isolated, so as to minimize stress on newly fledged birds. The sex of 20 and 35 dph birds could usually be distinguished by plumage, however, we routinely confirmed the sex of individuals by gonadal inspection. Birds were sacrificed by decapitation and their brains removed. For electrophysiology experiments brains were bisected along the midline, immersed in the ice-cold cutting solution, and processed as described below. For a subset of these brains (N = 4 per sex per age group), we reserved one hemisphere for electrophysiology experiments and the other was placed in a plastic mold, covered with ice-cold Tissue-Tek OCT (Sakura-Finetek; Torrance, CA), and frozen in a dry ice/isopropanol slurry for in situ hybridization. The use of right vs. left hemispheres was balanced to account for any hemispheric differences in electrophysiology or gene expression. Both hemispheric sides were used for electrophysiological recordings.

**Isolation experiment.** Three clutches of zebra finches were initially reared with full access to their mother and father. At 20 (±2) dph the father was removed to prevent juvenile exposure to tutor song during the song memory acquisition phase. At 30 dph, we removed a single juvenile male from each clutch and housed it under complete social isolation that prevented all access to normal finch auditory and social cues[80,81]. We then reintroduced the father (tutor) to the original family cage, and all remaining siblings, including matched juvenile males, were reared normally. Socially isolated (n = 3) and tutored (n = 5) birds were reared until 65 dph. Over three consecutive days (63–65 dph), the female-directed song was recorded during the period from lights-on (10 AM) until noon (12 PM)[122]. Songs were analyzed using Sound Analysis Pro (SAP) software[123]. Isolated (n = 3) and Tutored (n = 5)

males were then sacrificed and their brains processed for electrophysiology recordings as described above for all other animals. To quantify the degree of tutor song imitation we randomly selected 9–10 song bouts from each tutor and tutee pair (n = 2) and used Raven Lite 2.0.1[124] to isolate the first motif from each bout. We then used SAP with default parameters to perform all possible bidirectional pairwise spectral comparisons within each tutor/tutee pair (n = 192 per pair) and calculated a mean similarity score based on the "accuracy" measurement. A similar analysis was not performed between isolated birds and their fathers because their songs shared no obvious resemblance (Supplementary Fig. 8a, top vs. bottom sonogram). To quantify the level of stereotypy, we used a unidirectional pairwise analysis of 8–10 motifs produced by each bird (n = 41–45 comparisons per bird). We then averaged the values separately for the tutored and isolated birds and compared the two groups with a standard *t* test.

**In situ hybridization.** To compare mRNA expression levels for *SCN3B* and *SCN4B* across developmental ages and between sexes, brains sections (thickness = 10 μm) were cut on a cryostat onto glass microscope slides (Superfrost Plus; Fisher Scientific) and stored at −80 °C. For each brain, a set of slides consisting of every 10th slide was fixed and stained for Nissl using an established protocol. Slides were examined under dark- and bright-field microscopy to identify sections containing the core region of nucleus RA, i.e., the sections where RA appears largest. In situ hybridizations were conducted using established protocols[125] and were performed in separate batches to standardize fixation and to establish optimal hybridization conditions for each gene. Slides were hybridized under pre-optimized conditions with DIG-labeled riboprobes synthesized from BSSHII digested cDNA clones obtained from the ESTIMA: Songbird clone collection[126], corresponding to GenBank entries FE734016 (*SCN3B*; Navβ3), FE730991 (*SCN4B*; Navβ4), and DV957065 (*SCN1B*; Navβ1). After hybridization, slides were washed, blocked, incubated with alkaline phosphatase-conjugated anti-DIG antibody (Roche, #110932749100, 1:600), and developed overnight in BCIP/NBT chromogen (Perkin Elmer). All slides for a given probe were run together using standardized and pre-established optimal conditions (i.e., wash times and temperatures, amount of probe, the concentration of chromogen, duration of incubation in detection solution). We note that we routinely include no-probe negative control slides in our hybridizations, as well as slides hybridized with a strongly expressed gene (*GAD2*) whose expression is constitutively higher than that of the ion channel genes under study, and we control tightly the exposure time in chromogen so as prevent signal saturation compared to this more highly expressed control gene. Slides were coverslipped with VectaMount (Vector) permanent mounting medium, and then digitally photographed at 10× under brightfield illumination with a Lumina HR camera mounted on a Nikon E600 microscope using standardized filter and camera settings. Images were stored as TIFF files and analyzed further using the FIJI distribution of ImageJ[127]. We note that high-resolution images of *SCN1B*, *SCN3B*, and *SCN4B* expression in the adult male zebra finch brain are available on the Zebra Finch Expression Brain Expression Atlas (ZEBrA; www.zebrafinchatlas.org)[128]. The evaluation of probe specificity, as well as the sequence comparisons and estimates of percent identity between avian and mammalian orthologs of ion channel genes, were performed through BLAST and Clustal analyses[52,99,128].

**Image analysis and expression quantification.** For each image, we quantified both expression levels based on labeling intensity (i.e., average pixel intensity) and the number of cells expressing mRNA per unit area. Since the density of cells in RA is known to change during development and to vary by sex[61], we first estimated the number of neurons in RA and caudal to RA for each bird by placing a 200 × 200 μm window over target areas in the images of the Nissl-stained sections containing the largest RA per bird. We next measured the average pixel intensity (scale: 0–256) in an identical 200 × 200 μm window placed over each target area in the images of hybridized sections adjacent to the Nissl-stained sections. From the values obtained over target areas of interest we then subtracted an average background level measured over an adjacent control area in the intermediate arcopallium that was deemed to have no mRNA expression. Finally, we divided the module of the background-corrected pixel intensity value by the number of Nissl-counted cells in order to obtain a measurement of the average pixel intensity per cell. We also quantified the number of labeled cells in each arcopallial region by first establishing a threshold of expression 2.5× above the background level. Standard binary filters were applied and the FIJI "Analyze Particles" algorithm was used to count the number of labeled cells per 200 μm². This value was further normalized for comparisons across different ages and sex by dividing by the number of Nissl-labeled cells from the adjacent section. We note that for all quantified ion channel gene probes our optimized hybridization conditions resulted in levels of non-specific background (density measurements taken over tissue regions lacking cells) that were essentially indistinguishable from the no-probe control hybridizations. Under these optimized conditions, the densitometric values were consistent and monotonic with labeled cell counts and were not saturated relative to the more highly expressed control probe (*GAD2*).

**Slice preparation for electrophysiology experiments.** Sagittal (200 μm thick) and frontal (150 μm thick) slices were cut on a vibratome slicer (VT1000, Leica) in

an ice-cold cutting solution containing (in mM): 119 NaCl, 2.5 KCl, 8 MgSO$_4$, 16.2 NaHCO$_3$, 10 HEPES, 1 NaH$_2$PO$_4$, 0.5 CaCl$_2$, 11 D-Glucose, 35 Sucrose, pH 7.3–7.4 when bubbled with carbogen (95% O$_2$, 5% CO$_2$; osmolarity ~330–340 mOsm). Slices were then transferred to an incubation chamber containing artificial cerebral spinal fluid (aCSF) with (in mM): 119 NaCl, 2.5 KCl, 1.3 MgSO$_4$, 26.2 NaHCO$_3$, 1 NaH$_2$PO$_4$, 1.5 CaCl$_2$, 11 D-Glucose, 35 Sucrose, pH 7.3–7.4 when bubbled with carbogen (95% O$_2$, 5% CO$_2$; osmolarity ~330–340 mOsm) for 10 min at 37 °C, followed by a room temperature incubation for ~30 min prior to start of electro-physiology experiments. For low-sodium aCSF experiments (see Fig. 5), 119 NaCl mM extracellular sodium was replaced with 30 mM NaCl and 90 mM NMDG, the pH was adjusted to 7.4 with HCl, and the osmolarity was adjusted to ~330–340 with sucrose.

**Patch-clamp electrophysiology**. RA could be readily visualized in males and females of all ages via infrared differential interference contrast microscopy (IR-DIC) (Supplementary Fig. 5). Whole-cell CC recordings were performed at room temperature (~24 °C) unless otherwise indicated. For experiments performed at 40 °C, the bath solution was warmed using an in-line heater (Warner Instruments, Hamden, CT). The temperature for these experiments varied up to ±2 °C. Slices were perfused with carbogen-bubbled aCSF (1–2 ml/min) and neurons were visualized with an IR-DIC microscope (Zeiss Examiner.A1) under a 40× water immersion lens coupled to a CCD camera (Q-Click; Q-imaging, Surrey, BC, Canada). Whole-cell VC and CC recordings were made using a HEKA EPC-10/2 amplifier controlled by Patchmaster software (HEKA, Ludwigshafen/Rhein, Germany). Data were acquired at 40 kHz and low-pass filtered at 2.9 kHz. Patch pipettes were pulled from standard borosilicate capillary glass (WPI, Sarasota, FL) with a P97 puller (Sutter Instruments, Novato, CA). All recording pipettes had a 3.0 to 6.0 MΩ open-tip resistance in the bath solution. Electrophysiology data were analyzed offline using custom-written routines in IGOR PRO (WaveMetrics, Lake Oswego, OR).

For VC recordings, intracellular solutions contained the following (in mM): 142.5 Cs-Gluconate, 11 CsCl, 5.5 Na$_2$-phosphocreatine, 10.9 HEPES, 5.5 EGTA, 10.9 TEA-Cl, 4.2 Mg-ATP, and 0.545 GTP, pH adjusted to 7.3 with CsOH, ~330–340 mOsm. Alexa fluor 488 hydrazide (Thermofisher; 10 μM) was added to this solution to aid in cell-type morphology discrimination via epifluorescent microscopy (X-Cite series 120, Excelitas Technologies, Waltham, MA). In recordings where the series resistance ($R_s$) was compensated to 5 MΩ, the average uncompensated $R_s$ = 10.2 ± 0.5 MΩ ($n$ = 66 cells for male/female). In recordings in Fig. 5 where $R_s$ was compensated to 1 MΩ, the uncompensated $R_s$ = 7.2 ± 0.5 MΩ ($n$ = 8 for males at 20 dph and adults). In order to isolate Na$^+$ currents, slices were exposed to bath applied CdCl$_2$ (100 μM), 4-AP (100 μM), Picrotoxin (100 μM), TEA (10 mM), CNQX (10 μM), and APV (100 μM) for ~5 min prior to running VC protocols. After protocols were applied, tetrodotoxin (TTX; 1 μM) was bath applied and the same protocols were repeated after Na$^+$ currents were eliminated. Na$^+$ currents were isolated by subtracting the TTX-insensitive current traces from the initial traces. Capacitive currents generated during VC recordings were eliminated by P/4 subtraction. Recordings were not corrected for a measured liquid junction potential of +12 mV for 119 mM NaCl and +13 mV for 30 mM NaCl containing extracellular solutions.

For CC recordings, intracellular solutions contained (in mM): 142.5 K-Gluconate, 21.9 KCl, 5.5 Na$_2$-phosphocreatine, 10.9 HEPES, 5.5 EGTA, 4.2 Mg-ATP, 0.545 GTP, and 10 μM Alexfluor 488 hydrazide, pH adjusted to 7.3 with KOH, ~330–340 mOsm. Synaptic currents were blocked by bath applying Picrotoxin (100 μM), APV (100 μM), and CNQX (10 μM) (Tocris Bioscience) for ~3 min prior to all recordings. To initiate CC recordings, we first established a gigaohm seal in the VC configuration, set the pipette capacitance compensation (C-fast), and then set the voltage command to −70 mV. We then applied negative pressure to break into the cell. Once stable, we switched to the fast CC configuration. Experiments in CC were carried out within a 15-min period. We noted that the resting membrane potential tended to hyperpolarize to the same degree (~10 mV) in both juveniles and adults during these CC recordings[129]. Recordings in which the resting membrane potential deviated by >15 mV were discarded. We note that recordings were not corrected for a measured liquid junction potential of 9 mV.

Estimated current clamp measurements of membrane capacitance ($C_m$) were calculated from the measured membrane time constant ($\tau_m$) and input resistance ($R_{in}$) using the equation:

$$\tau_m(ms) = R_{in}(M\Omega) . C_m(pF) \quad (3)$$

Estimated peak Na$^+$ current ($I_{Na}$) activated during the upstroke of the AP was calculated from the capacitive membrane current using the equation:

$$max . I_{Na}(pA) = C_m(pF) . max . \frac{dV_m(V)}{dt(s)} \quad (4)$$

where peak depolarization rate is max. $dV_m/dt$ (see Vaaga et al.[73] and Yu et al.[49]).

Dialyzed peptides: Mouse β4-WT (KKLITFILKKTREK) and β4-Scr (KIKIRFKTKTLELK) peptides (described in Grieco et al.[55]) were synthesized by Thermofisher scientific at 98% purity and diluted to 100 μm in internal solutions for CC and VC experiments. The mouse peptide was used because, compared to the finch (and chicken) peptide, it was found to be more readily soluble in our

internal solutions[57]. Importantly, the finch peptide is highly similar to peptides derived from other species known to produce $I_{NaR}$. The finch peptide shows 87% and 93% conservation (conserved amino acid substitutions included) to the mouse and human peptides, respectively. Furthermore, the finch and chicken peptides show 93% identity (14/15 identical residues), and all three species (finch, chicken, and mouse) share preservation of the crucial phenylalanine at position 6 and lysines at positions 9 and 10.

Dynamic clamp parameters: Dynamic clamp was implemented in a Hewlett Packard PC (Z600 workstation) running Windows 10 software[130]. Voltage values were read and calculated currents outputted through a National Instruments DAQ board (PCIe-6321) controlled by Igor Pro 8 using the NIDAQ Tolls MX package (Wavemetrics). We utilized a model deposited in ModelDB within the *Senselab* database (see accession number 127021)[85,131]. The $I_{NaR}$ current was modeled as a single gate relying on two particles: one for activating (s) and one for inactivating (f). Each of these particles has a separate probability for opening (i.e., going to a permissive state: α) and closing (i.e., going to a non-permissive state: β). In order to match the kinetics of the model to zebra finch $I_{NaR}$ currents, we made minor changes to the equations describing the rate constants in the original model[85]. We adjusted two parameters of the s particle rate constants: (1) an exponential term denominator ($K\alpha_s$ from −6.82 to −1.0) and (2) the magnitude of the multiplying term β rate constant ($A\beta_s$ from 0.0156 to 0.05).

To calculate the in silico current, we used a set of first-order Hodgin–Huxley type equations:

$$I_{Nar} = g_{Nar} * s * f * (V − E_{Nar}) \quad (5)$$

$$\tau_x = \frac{1}{(\alpha_x + \beta_x)} \quad (6)$$

$$x^\infty = \alpha_x * \tau_x \quad (7)$$

$$\frac{dx}{dt} = \frac{x^\infty − x}{\tau_x} \quad (8)$$

Where $x$ represents either the $s$ or $f$ gating particles. The differential Eq. (8) was solved using a first-order Euler method[132] with a time step of 0.03 ms. The $g_{Nar}$ represents the maximum conductance (in mS) and the calculated reversal potential ($E_{Nar}$) was +67 mV. All neurons were kept approximately at −70 mV throughout the CC recordings. To trigger APs, a step current injection of +300 pA (relative to the holding current) was applied for 1 s. A complete list of the numeric parameters and equations is available in ModelDB (modeldb.yale.edu/127021)[85,131].

**Statistical analysis**. Electrophysiology data were analyzed using standard macros available with Igor Pro v8.0 (Wavemetrics). In situ hybridization was analyzed using the Anaconda v1.9.7 distribution of Python v3.4 with a Jupyter environment v6.0.1. Electrophysiology and gene expression data were further analyzed in Excel (Microsoft Office 2016), and statistical analyses were performed in Prism v8.3.0 (GraphPad). Songs were analyzed using Sound Analysis Pro (SAP) software v2011.087 (soundanaylsispro.com)[123] and Raven Lite v2.0.1 (ravensoundsoftware.com)[124]. Means and SE are reported unless otherwise noted.

**Reporting summary**. Further information on research design is available in the Nature Research Reporting Summary linked to this article.

## Data availability

Source data for Figs. 2–10 and Supplementary Figs. 1, 3, and 6–11 are provided as a source data file. A reporting summary for this article is available as a Supplementary Information file. Source data are provided with this paper.

## Code availability

The dynamic clamp model generated in this study has been deposited in the Senselab database under ModelDB, accession code 267132.

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

## Acknowledgements

We would like to thank Samatha Friedrich for their help with zebra finch breeding and Dr. Pepe Alcami for instructive discussions and suggestions. This work was funded by the following grants: NSF1456302, NSF1645199, GM120464, DC004274, and DC012938.

## Author contributions

B.M.Z., P.V.L., C.V.M. and H.v.G. designed the research. B.M.Z., A.A.N. and A.D. performed the research, B.M.Z., A.A.N., A.D., P.V.L., C.V.M. and H.v.G. analyzed the data, B.M.Z., P.V.L., C.V.M. and H.v.G. wrote the paper.

## Competing interests

The authors declare no competing interests.
