## [Peer Review File · Nature Communications]

Resurgent Na⁺ currents promote ultrafast spiking in projection neurons that drive fine motor controlREVIEWER COMMENTS

Reviewer #1 (Remarks to the Author):

The current manuscript by Zernel et al. leverages a molecular finding into a set of experiments investigating spike properties of a motor area in the songbird singing circuit. There are many strengths of this manuscript. It is well written, with detailed technical descriptions. It presents an intriguing set of results with a successful blend of molecular and physiological measures. It makes a leap forward from the initial observation of Nav-beta-4 mRNA in the motor nucleus RA to properties of cells that may relate to the highly stereotyped and finely-timed behavior of song. It furthers the literature on functional similarities in brain function across species, making the findings of interest to motor system researchers, songbird researchers, and those interested in brain evolution.

There is, however, one main component missing: behavior. A link between behavior and RA and Nav-beta-4, would significantly elevate the impact of the rest of the data. As song stereotypy appears to be the characteristic of song that would be most related to Nav-beta-4 channels, there may be several options including song-isolated males who have less song stereotypy and deafened adult males who show deteriorations in song structure. Recording the birds prior to sacrifice for brain metrics would allow for individual-based analysis.

Sexual differentiation of RA volume is not simply a loss of cells in females, as portrayed (line 370), which is important because it requires consideration of sex differences in the local cell environment that may or may not be inter-related with Nav-beta-4 abundance.

The discussion in most cases nicely provides context for the data but is disproportionately long and information provided is not always clearly relevant to the direct implications of the results.

Minor points:

- "mRNA expression" is redundant
- including informative headers for all subsections may help guide the reader through main points
- having colored bar legends within the figures (graphs) would be convenient

Reviewer #2 (Remarks to the Author):

This is a complex study that explores the molecular mechanisms that contribute to differential excitability of zebra finch song bird motor neuron activity. In particular, the high frequency activity of projection neurons in males is investigated. Firing in these neurons increases with changes in the ability of the adult male to sing. The role of Nav beta-4 subunits is explored. This system has interesting parallels to mammal high frequency layer 5 pyramidal projection neurons. The data are intriguing and generally of high quality and rigor. The differences between males and females, age dependent changes and dynamic clamp experiments are especially impressive. Overall an exciting study that will attract significant interest. However, there are several that should be addressed to improve the validity of the conclusions and the overall presentation.

Page 6, line 242: The use of this type of fit for onset of current kinetics needs to be better justified. A simple m3h type Hodgkin-Huxley fit often does a very good job of measuring activation kinetics. The $N=3$ can be varied to better determine the activation "gates". Sodium channel inactivation can impact the apparent fit to the activation of the currents and thus it is difficult to interpret the meaning of an $N=10.7$ versus $N=3.7$ when not taking into account inactivation kinetics.

The study uses a SCN4B peptide to explore if lack of substantial Nav-beta4 protein might contribute to the differences in excitability observed between 20 dph and adult RAPNs. While the methods state that mouse peptide is used as there is high conservation, the mouse and corresponding zebra finch peptide (KRVVLFIIKKTQDGK) are less than 50% conserved. This raises

concerns about the usefulness of the peptide experiments. KRVLFIKKKTQDGK

Line 329: It is curious that Beta4-WT peptide broadens the action potential. Peptide binding should increase the apparent rate of inactivation and thus might be expected to narrow the AP. Are resurgent currents kicking in to broaden the action potential? If so, why are these not observed in adults with large resurgent currents? Perhaps the mammalian peptide has distinct kinetics. As is briefly mentioned, Kv channels are likely to be very important in the action potential dynamics. Do Kv3 channel expression change with development in the male finches? Do RAPNs express FGF14? This could also be a factor in AP dynamics that should at least be better discussed.

Line 305: temperature is misspelled.

Page 530: the wording is a bit awkward. Maybe "which bind the Nav alpha subunit non-covalently"?

Given that figure 1 uses different temperatures, it might help if for figure 2 the temperature used was stated. Maybe around line 136 or in the figure legend.

Reviewer #3 (Remarks to the Author):

In their paper, Zemel et al. examined the intrinsic properties of the projection neurons in the zebra finch song nucleus RA. They found that the spiking properties of these neurons greatly mimic those of specialized pyramidal neurons in mammalian motor cortex (L5PNs), and they arise throughout the critical period of vocal learning in males accompanied with expression of voltage-gated Na⁺ channel auxiliary beta4 subunit (Nav beta4) responsible for producing a large resurgent sodium current (INaR). This enables high-frequency spike burst firing that is required for the superfast, temporally precise and refined control of musculature involved in singing.

This is a thorough study that combines experimental and computational approaches with sections that are clear and easy to follow. The findings in each section are interesting on their own, but a strength of the study is to combine an electrophysiological approach (intracellular recordings from RA projection neurons, termed RAPNs), and a molecular one (quantification of sodium ion channel transcripts) to demonstrate the high plasticity of intrinsic excitability while evaluating this considering sex differences and song developmental stages. These all add to a comprehensive study whose principal results are secure. The beautiful and compelling histology certainly adds to this sense of confidence.

There are also a few global weaknesses. Perhaps the strongest weakness of the paper is related, that the results while compelling do not use the mechanistic insights when evaluating the various players in this study (e.g. resurgent Na⁺ currents, or RAPN activity) to alter our understanding of brain function. To a tangible degree this reduces the impact of the paper.

Another weakness will be easier to address. There is a confusion in this paper between the concepts of temporal precision and temporal coding in spike timing. The paper investigates the temporal precision of RA projection neuron spike timing. Temporal coding refers to a hypothesis about the relative contribution of timing (in contrast to rate) of spike firing, that is the window of integration that carries information. The paper does not provide data to evaluate the contribution of temporal coding to the activity of RAPNs, so the use of the term is misleading. The result that resurgent currents are the mechanism that supports the high temporal precision of RAPNs is worthwhile, but that temporal precision has been well known for about 25 years now, and its role in temporal coding has been discussed in several publications. This manuscript does not contribute in a substantial way to that discussion. For example, line 22 (Abstract) "and are well suited for precise temporal coding". This is correct but includes a hidden hypothesis. More accurately, say "and yield temporal precision of spike timing" which is hypothesis neutral.

There are also some other comments:

The authors claim that their results provide strong evidence linking the presence of Nav^{β4} to large

resurgent Na⁺ currents (INaR) for the first time, but this link is previously established and shown to contribute to excitability of the sensory neurons of the dorsal root ganglion (DRG) (Barbosa et al., 2015). In addition, when expression of NaV beta4 is knocked down by siRNA in cultured cerebellar granule cells, resurgent current is reduced and repetitive firing is compromised establishing NaV beta4 as a good candidate for an endogenous open channel blocker (Lewis and Raman, 2014). Increased intrinsic membrane excitability linked to resurgent Na current and NaV beta4 have also been revealed in the medial entorhinal cortex associated with temporal lobe epilepsy (TLE).

One of the interestingly new result supported by the authors is that the intrinsic excitability of RAPNs undergoes substantial developmental changes throughout the period of vocal learning. However, the degree to which these age-dependent changes are driven by motor practice and learning vs being genetically predefined is not examined or discussed. This result; nonetheless, constitutes a baseline against which the effect of targeted manipulations whether behavioral, pharmacological, or others can be compared in the future to further define the factors involved in the developing song system.

Not all cells with resurgent current express detectable NaV beta4. Examples of such cells include the GABAergic neurons of the medial vestibular nucleus (Kodama et al. 2012) and the hippocampal dentate granule (DG) neurons (Castelli et al. 2007). Hence, referring to the beta 4 subunit as a 'molecular requirement' (line 431) is inaccurate.

While it seems that the resurgent current is closely related to the distinctive firing behavior of RAPN, especially the ability to fire repetitive action potentials, the contribution of other ion currents (other than the transient sodium current) which in turn influences spike shape and firing, was missing. Developmental changes may also occur to the entire complement of ion channels present in these neurons, thus contributing to the variations in the firing phenotype of RAPN seen through development. Such a study for example exploring pharmacological manipulations of RAPN is beyond the scope of the present paper, but as written these limitations are overlooked.

In the Discussion the involvement of potassium channels to the repetitive and narrow spike firing phenotype is discussed, while the Results section imply that the resurgent current is solely promoting the narrow action potentials (lines 256-257). Indeed, presence of the resurgent current affects spike morphology, however, more studies are needed to prove that the resurgent current is solely responsible for this behavior. The Results section needs to be modified accordingly.

lines 195-213 are confusing. What is the significance of examining the developmental regulation of subunit 1 given that (in contrast to subunit 4) its function is not known/not stated by the authors? The authors come to the conclusion at the end of this section that subunit 4 is correlated with the increase in resurgent sodium current throughout vocal learning development. What is then the role of subunit 1? Is it also developmentally regulated but by a different behavior? Can its role be omitted? What if the role of subunit 4 that is facilitation of high frequency firing seen in mammals does not apply in songbirds?

Minor:

Line 78: We provide "strong" evidence linking
"extensive" might be better if you feel a need for a qualifier

Line 141: "These results reveal an exceptionally large INaR in RAPNs compared to those recorded in mammalian and chicken brainstem neurons"
provide some summary quantification here to help the reader

Line 256: The persistent Na⁺ current mentioned is not referred to in the indicated figure (Fig. 5h).

Line 434: "well suited" better than well "engineered"?

Line 480: Sentence starting "Taken together..." is confusing. The intrinsic properties of RAPNs differ from those of HVCPNs, but what about those intrinsic property differences represent a

"starkly contrasting" difference? Also, since HVC is thought to be a site of temporal coding, how does the sentence make sense?

Line 496: spelling: "Deveopmental"

indicate IR-DIC in Supp 5 legend

REVIEWER COMMENTS

Reviewer #1 (Remarks to the Authors):

The current manuscript by Zernel et al. leverages a molecular finding into a set of experiments investigating spike properties of a motor area in the songbird singing circuit. There are many strengths of this manuscript. It is well written, with detailed technical descriptions. It presents an intriguing set of results with a successful blend of molecular and physiological measures. It makes a leap forward from the initial observation of Nav-beta-4 mRNA in the motor nucleus RA to properties of cells that may relate to the highly stereotyped and finely-timed behavior of song. It furthers the literature on functional similarities in brain function across species, making the findings of interest to motor system researchers, songbird researchers, and those interested in brain evolution.

There is, however, one main component missing: behavior. A link between behavior and RA and Nav-beta-4, would significantly elevate the impact of the rest of the data. As song stereotypy appears to be the characteristic of song that would be most related to Nav-beta-4 channels, there may be several options including song-isolated males who have less song stereotypy and deafened adult males who show deteriorations in song structure. Recording the birds prior to sacrifice for brain metrics would allow for individual-based analysis.

We thank the reviewer for the positive overall assessment of our study and the helpful suggestions. To address the reviewer's request to include behavior, we have now added new data comparing the recorded songs and the excitable properties of RAPNs between 65 dph juveniles that were raised in isolation (new Supplemental Fig. 8) and their siblings raised with normal exposure to a conspecific tutor song within a family context. As expected, the songs of tutored birds closely resembled the songs of their tutors, indicative of vocal imitation. In contrast, the songs of isolates had low similarity to the tutors' (their fathers') songs to which they were not exposed, consistent with previous reports (e.g., Price, *J. Comp Phys Psych* 1979; Morrison and Nottebohm, *J. Neurobio.*, 1993; Kojima and Doupe, *Euro. J. Neurosci.*, 2008). Importantly, despite obvious differences in song similarity, the major excitable properties of RAPNs, including spike shape and evoked firing properties, were indistinguishable between tutored and isolate birds. Thus, the maturation of RAPN excitable properties is clearly not dependent on exposure to a tutor song or the ability to copy that song.

We note that we chose to examine 65 dph juveniles because we needed to study an age where we could measure song imitation in tutored birds and the effects of isolation on the song of isolated birds. Furthermore, adding the 65 dph data extends the examined developmental age range in our study (previously 20, 35, 50 dph juveniles) noting that 65 dph is within a period of very active auditorily-guided vocal learning.

Interestingly, the stereotypy index did not differ significantly between the tutored and isolate groups at 65 dph, noting the considerable variability in both groups. We also note that the most marked changes in intrinsic excitable properties of RAPNs occur between 20 and 50 dph. Thus, they overlap with the period when the juvenile birds produce increasingly more complex syllable patterns, as they transition through the early phases of song development (subsong and early plastic song), rather than the late phases of song imitation and crystallization.

The new findings are presented in a new Supplemental Figure 8. The experimental details and analyses have been inserted in Methods and Results, with comments on their significance in the Discussion. Please see lines 79-80, 322-342, 545-552 and 663-680. We acknowledge that further studies are needed to fully understand the relationship between molecular and excitable properties of RAPNs and song behavior. Our study, however, is the culmination of an extensive and dedicated effort and contributes substantial insights into molecular, biophysical and physiological features of RAPNs. In its revised form, it now also provides new information on important behavioral variables (social contact and vocal imitation). Further manipulations (deafening, lesioning, hormone treatment, etc.) to study the determinants RAPN excitable properties would require extensive additional effort

that we believe is beyond the scope of an already extensive and detailed study (10 multi-panel Figures, 11 Supplementary Figures and 4 Tables).

Sexual differentiation of RA volume is not simply a loss of cells in females, as portrayed (line 370), which is important because it requires consideration of sex differences in the local cell environment that may or may not be inter-related with Nav-beta-4 abundance.

We agree with the reviewer that the developmental decrease in volume of the female zebra finch RA is not simply due to loss of cells, but also involves changes in cell size, dendritic arborizations, synaptic connectivity, and other parameters, and is likely due to complex interactions among sex hormones and their receptors, growth factors, sex-chromosome genes, and possibly other still unknown factors. Whether or not sex differences in locally acting factors relate to Nav β 4 expression is unknown, but represents an important question for future studies. We have modified the relevant section to more accurately address the reviewer's concern (see line 402-404 in the Results and 617-619 in Discussion).

The discussion in most cases nicely provides context for the data but is disproportionately long and information provided is not always clearly relevant to the direct implications of the results.

To address this concern, we have modified several portions of the Discussion so that it more closely addresses direct implications of our results. All major changes are marked in red. Please see the new Discussion in pages 10, 11 and 12 where we explain some of the major insights of our results.

Minor points:

- "mRNA expression" is redundant

We have eliminated the term "mRNA" where appropriate to avoid this redundancy.

- including informative headers for all subsections may help guide the reader through main points

We previously had 8 headers for the Results section and 8 headers in the Discussion. However, in order to comply with reviewers' requests, we have added a new Results subsection and header describing the social isolation experiment, and revised four headers of the Discussion so that the subsection more clearly reflects the content discussed. Please see lines 322, 464, 477, 530 and 581.

- having colored bar legends within the figures (graphs) would be convenient

We have included additional colored bar legends where relevant. Please refer to Fig. 2, 3, 4, 5 and 6.

Reviewer #2 (Remarks to the Author):

This is a complex study that explores the molecular mechanisms that contribute to differential excitability of zebra finch song bird motor neuron activity. In particular, the high frequency activity of projection neurons in males is investigated. Firing in these neurons increases with changes in the ability of the adult male to sing. The role of Nav beta-4 subunits is explored. This system has interesting parallels to mammal high frequency layer 5 pyramidal projection neurons. The data are intriguing and generally of high quality and rigor. The differences between males and females, age dependent changes and dynamic clamp experiments are especially impressive. Overall an exciting study that will attract significant interest. However, there are several issues that should be addressed to improve the validity of the conclusions and the overall presentation.

We thank the reviewer for the overall very positive assessment of the study and detailed suggestions for improving the analysis and presentation.

Page 6, line 242: The use of this type of fit for onset of current kinetics needs to be better justified. A simple m3h type Hodgkin-Huxley fit often does a very good job of measuring activation kinetics. The N=3 can be varied to better determine the activation “gates”. Sodium channel inactivation can impact the apparent fit to the activation of the currents and thus it is difficult to interpret the meaning of an N=10.7 versus N=3.7 when not taking into account inactivation kinetics.

We acknowledge that we cannot fully discount the effects of rapid inactivation, however we believe the N values obtained are useful to quantify the kinetics and also for comparison of adult and juvenile finch Na⁺ currents, as they were obtained using the same exact method across ages. They are also useful for comparisons to other CNS neurons. A similar analysis was used for layer V pyramidal neurons (Almog et al., 2018; N = 5 for +30 mV steps) showing that inactivation contributed little to a fit to 50% of the peak current. We used a reduced range, fitting to 40% of our peak I_{NaT} to further limit this contribution. Similar analysis was also done by Magistretti et al. (2006) in cerebellar granule cells, where N = 9 for +10 mV steps. We note that the data in juveniles are indeed close to what is expected for a typical m3h Hodgkin and Huxley fit, whereas the adult data are similar to that of mammalian layer V projection neurons (Almog et al., 2018), which have been described to depart from a simple m3h Hodgkin and Huxley fit. We have noted the caveat regarding inactivation in the text. Please see line 251-253.

The study uses a SCN4B peptide to explore if lack of substantial Nav-beta4 protein might contribute to the differences in excitability observed between 20 dph and adult RAPNs. While the methods state that mouse peptide is used as there is high conservation, the mouse and corresponding zebra finch peptide (KRVVLFIIKKTQDGK) are less than 50% conserved. This raises concerns about the usefulness of the peptide experiments. KRVVLFIIKKTQDGK

The reviewer raises an important issue that requires clarification. While the percent identity of this peptide is ~47% (6/15 identical residues) between finch and mouse, the majority of substitutions (6) are conservative (same class of residues). Thus, the overall level of conservation between finch and mouse peptides can be considered as being 87% (and even higher - 93% - compared to human). Furthermore, the finch and chicken peptides show 93% identity (14/15 identical residues), and all three species (finch, chicken and mouse) share a critical phenylalanine at position 6 and lysines at positions 9 and 10 thought to be important for the transient block of Nav channels (Lewis and Raman, 2014).

A previous study (Lewis and Raman, J. Neurosci., 2011) describes difficulty with solubilization of the chicken peptide due to polar residues that are also present in the finch peptide. While the strategy of extending this peptide enabled solubilization of the chicken peptide in that study, we were unable to solubilize the extended zebra finch peptide in our internal solution, thus it was not practical to use the finch peptide in our studies.

Importantly, the peptides from both mouse and chicken produce resurgent currents with similar kinetics when delivered to neurons originally lacking this current. Given the high conservation between the finch and chicken peptides, and the previous evidence that the latter is associated with resurgent currents, it is reasonable to assume a similar conserved function for the finch peptide. We thus think that the more soluble mouse peptide provides a useful tool for examining the potential link of Navβ4 expression to resurgent current regulation in zebra finch neurons. We have modified the language in the Methods to more accurately reflect the sequence similarities between the zebra finch, chicken and mouse peptides. Please see lines 797-804.

Line 329: It is curious that Beta4-WT peptide broadens the action potential. Peptide binding should increase the apparent rate of inactivation and thus might be expected to narrow the AP. Are resurgent currents kicking in to

broaden the action potential? If so, why are these not observed in adults with large resurgent currents? Perhaps the mammalian peptide has distinct kinetics.

The reviewer touches on another important point. Indeed, we think the resurgent current is activating during the repolarization of the juvenile finch AP. This is what may initiate the “spikelet” seen prior to a full repolarization of the 1st AP. We believe the reason for AP broadening is two-fold:

First, the juvenile AP is more than 2x as broad as the adult (Table 1). This prolonged duration at depolarized voltages, coupled with slowly inactivating Navs (Fig. 5) allow for the peptide to block during prolonged depolarized potentials, and unblock, promoting a resurgent current during prolonged AP repolarization.

Second, the adult has a more than 4x larger maximum rate of repolarization during the AP (Table 1). Preliminary evidence we have gathered for a follow-up study strongly suggests that in contrast to juveniles, adult RAPNs express high threshold Kv3.x channels. The short adult AP duration due to rapid repolarization may not be significantly modulated by the relatively slower activation of the resurgent current, which we propose is likely more involved in facilitating threshold for subsequent spikes during interspike periods. We discuss this as it relates to the “spikelet” in lines 603-605 and 609-610.

As is briefly mentioned, Kv channels are likely to be very important in the action potential dynamics. Do Kv3 channel expression change with development in the male finches?

As mentioned above, we have preliminary data suggesting RA specific, developmental upregulation of Kv3.1 mRNA in male zebra finches, but fully establishing this point is beyond the scope of the present study. We are currently working on experiments for a follow-up manuscript detailing the contribution of Kv3.1 currents to intrinsic excitable properties of neurons in the finch arcopallium (including RA). Lines 507-510.

Do RAPNs express FGF14? This could also be a factor in AP dynamics that should at least be better discussed.

We are certainly interested in exploring the expression of other putative transient blockers of Nav channels in RA. Interestingly, a previous study done by the Mello lab indicates that FGF14 is non-differentially expressed between RA and other parts of the zebra finch arcopallium outside RA (Lovell et al., BMC Genomics, 2018). Thus the expression of FGF14 does not correlate with the marked regional contrast seen in resurgent currents and Nav β 4 expression as shown in Figures 2, 3 and 4. We also ran *in situ* hybridization for FGF14 and verified very low expression within RA (data not shown), further arguing against a major role of FGF14 in resurgent currents in RA. We note that the probes used in our analysis cover all potential isoforms. We have added comments in the discussion addressing the role for FGF14 in producing resurgent currents. Please see lines 597-601.

Line 305: temperature is misspelled.

Thanks for the careful reading and editing. We have made the correction.

Page 530: the wording is a bit awkward. Maybe “which bind the Nav alpha subunit non-covalently”?

Thank you for the suggestion. We have made the change. See line 569.

Given that figure 1 uses different temperatures, it might help if for figure 2 the temperature used was stated. Maybe around line 136 or in the figure legend.

This is also a good suggestion, we have made the change in the text and figure legend. Please see lines 138 and 161 in the text and the legend of Figure 2 and 6.

Reviewer #3 (Remarks to the Author):

In their paper, Zemel et al. examined the intrinsic properties of the projection neurons in the zebra finch song nucleus RA. They found that the spiking properties of these neurons greatly mimic those of specialized pyramidal neurons in mammalian motor cortex (LSPNs), and they arise throughout the critical period of vocal learning in males accompanied with expression of voltage-gated Na⁺ channel auxiliary beta4 subunit (Nav beta4) responsible for producing a large resurgent sodium current (I_{NaR}). This enables high-frequency spike burst firing that is required for the superfast, temporally precise and refined control of musculature involved in singing.

This is a thorough study that combines experimental and computational approaches with sections that are clear and easy to follow. The findings in each section are interesting on their own, but a strength of the study is to combine an electrophysiological approach (intracellular recordings from RA projection neurons, termed RAPNs), and a molecular one (quantification of sodium ion channel transcripts) to demonstrate the high plasticity of intrinsic excitability while evaluating this considering sex differences and song developmental stages. These all add to a comprehensive study whose principal results are secure. The beautiful and compelling histology certainly adds to this sense of confidence.

We thank the reviewer for the highly positive overall assessment of the study.

There are also a few global weaknesses. Perhaps the strongest weakness of the paper is related, that the results while compelling do not use the mechanistic insights when evaluating the various players in this study (e.g. resurgent Na⁺ currents, or RAPN activity) to alter our understanding of brain function. To a tangible degree this reduces the impact of the paper.

We appreciate the reviewer's comment as it provided an opportunity for us to better highlight the main contributions of our study. Our data provide compelling new evidence that resurgent currents can regulate excitability in pyramidal-like cells located in cortical-like pallial areas of the vertebrate brain that are required for fine motor skills. This is an insight of considerable relevance, as previous evidence of resurgent currents in mammals stems primarily from examples in the peripheral nervous system, or in brainstem, cerebellum or subcortical areas, whereas the examples in pallial areas like hippocampus or entorhinal areas are less conclusive, or even questionable (as reviewed in Lewis and Raman, *J. Physiol.*, 2014). While setting an important precedent, we believe these findings will trigger strong interest in mammalian cortical physiologists and provide guidance for those wishing to verify the presence and characteristics of analogous currents in pyramidal cortical cells in rodents and non-human primates. In that regard, we note the high Navβ4 expression seen in deep cortical layers of rodents as shown in the Allen mouse atlas, for example. We have made changes to the Abstract (lines 25-26) and Introduction (lines 81-886), and reframed sections in the Discussion (lines 455-475 and 488-528) to highlight how our study lends important and compelling new insights into our current understanding of cortical brain function.

The reviewer's comment on brain function also led us to re-examine the implications of high frequency firing on the metabolic demands faced by RAPNs. In general, neurons that fire APs at high frequencies require more ATP to re-establish ionic concentrations via the Na/K ATPase. In mammals, this energy demand is mitigated by limiting 1) the duration at depolarized potentials by narrowing the AP waveform (Yu et al., *PLoS Comput. Biol.*, 2012) and 2) the overlap of Na⁺ and K⁺ conductances during the AP by ensuring rapid inactivation of I_{NaT} (Alle et al., *Science*, 2009; Fohlmeister, *Brain Res.*, 2009).

In our data set we were struck by the remarkable degree of AP narrowing (half-width = 0.18 ms in adults) while noting the increases in firing frequency of these neurons during development. We thus asked whether this narrow AP waveform, in addition to increasing temporal precision, also served to limit the time spent at depolarized V_m.

To answer this question, we further analyzed the membrane potential (V_m) during periods of evoked AP firing. To our surprise, we found that despite adult neurons firing significantly more evoked APs per unit time than juveniles, the average residency time of their membrane potentials is significantly more negative than values in juveniles. Although previous studies have described AP narrowing in L5PNs during development (Zhang et al. J., Neurophysiol., 2004), this analysis provided a novel way of evaluating the dynamic changes of V_m during evoked spiking across development. Importantly, these findings strongly suggest that like L5PNs, RAPNs possess intrinsic mechanisms to manage and mitigate the high metabolic cost of increases in spiking seen both in vitro and in vivo during vocal development. We believe this further contributes novel insights into brain function. We have thus added new panels to Fig. 6 (i and j) and modified the lines in the Results (287-294) and Discussion (514-528).

Another weakness will be easier to address. There is a confusion in this paper between the concepts of temporal precision and temporal coding in spike timing. The paper investigates the temporal precision of RA projection neuron spike timing. Temporal coding refers to a hypothesis about the relative contribution of timing (in contrast to rate) of spike firing, that is the window of integration that carries information. The paper does not provide data to evaluate the contribution of temporal coding to the activity of RAPNs, so the use of the term is misleading. The result that resurgent currents are the mechanism that supports the high temporal precision of RAPNs is worthwhile, but that temporal precision has been well known for about 25 years now, and its role in temporal coding has been discussed in several publications. This manuscript does not contribute in a substantial way to that discussion. For example, line 22 (Abstract) “and are well suited for precise temporal coding”. This is correct but includes a hidden hypothesis. More accurately, say “and yield temporal precision of spike timing” which is hypothesis neutral.

We appreciate the reviewer’s comments, and the need to distinguish between temporal precision of RAPNs firing and temporal coding. We also agree with the reviewer’s assessment that the identification of resurgent currents as a mechanism that supports high temporal precision, but not necessarily temporal coding, is a major contribution of the study. We have now changed the text to more accurately reflect that distinction and removed any reference to “temporal coding”. In addition, we now write in lines 467-468: “Low spike thresholds may facilitate sparse coding in HVC via the convergence of just a few spikes into a single RAPN.”

There are also some other comments:

The authors claim that their results provide strong evidence linking the presence of Nav-beta4 to large resurgent Na^+ currents (INaR) for the first time, but this link is previously established and shown to contribute to excitability of the sensory neurons of the dorsal root ganglion (DRG) (Barbosa et al., 2015). In addition, when expression of Nav beta4 is knocked down by siRNA in cultured cerebellar granule cells, resurgent current is reduced and repetitive firing is compromised establishing Nav beta4 as a good candidate for an endogenous open channel blocker (Lewis and Raman, 2014). Increased intrinsic membrane excitability linked to resurgent Na current and Nav beta4 have also been revealed in the medial entorhinal cortex associated with temporal lobe epilepsy (TLE).

We also appreciate the reviewer’s comments on this important issue. To clarify, we are well aware of these previous studies and indeed cited Barbosa et al., 2015 and other studies including a review (Lewis and Raman et al., J. Physiol., 2014), all of which pointing to a link between Nav β 4 and resurgent currents (Please see lines 131-132, 594). In addition, one of our labs was involved in showing the importance of resurgent currents for the excitability and high frequency firing of MNTB neurons in rodents (please see lines 582-585 and 605-607). Thus, it was never our intention to claim that the present study is the first to establish this link. However, we note that this link has been challenged by recent KO mice studies that have questioned the role of NavB4 in promoting resurgent current (White et. Al., J. Gen. Physiol., 2019). However, our paper lends strong support to the link between Nav β 4 and resurgent currents. We would also point to the fact that resurgent currents in RA are among the largest ever characterized (see lines 471-473), and the first such example in a songbird, but perhaps most importantly, they are the first conclusive evidence of such currents in a cortical motor circuit involved on fine motor skills.

One of the interestingly new result supported by the authors is that the intrinsic excitability of RAPNs undergoes substantial developmental changes throughout the period of vocal learning. However, the degree to which these age-dependent changes are driven by motor practice and learning vs being genetically predefined is not examined or discussed. This result; nonetheless, constitutes a baseline against which the effect of targeted manipulations whether behavioral, pharmacological, or others can be compared in the future to further define the factors involved in the developing song system.

We appreciate the reviewer's positive comments with regard to our demonstration of developmental changes in RAPN intrinsic excitability, and the indication that it will serve as a reference for future targeted manipulations to assess the contribution of genetic vs learning and vocal practice factors in modulating RAPNs' excitable properties. In response to a similar comment by Reviewer #1 (see reply to Reviewer #1, point 1 above), we have now included new results from 65 dph juveniles reared in isolation vs siblings reared in the presence of a conspecific tutor. The new data are now presented in Supplemental Fig 8, and discussed in lines 79-80, 322-342, 545-552 and 663-680; The data show no group differences in RAPN excitable properties at an age where there are striking differences in their songs, suggesting that the excitability changes we have described are not directly related to vocal imitation of the tutor song. We believe this is a significant addition that brings further insights to the study and helps guide future behavior efforts.

Not all cells with resurgent current express detectable Nav beta4. Examples of such cells include the GABAergic neurons of the medial vestibular nucleus (Kodama et al. 2012) and the hippocampal dentate granule (DG) neurons (Castelli et al. 2007). Hence, referring to the beta 4 subunit as a 'molecular requirement' (line 431) is inaccurate.

We agree that Navβ4 is not the only possible mechanism that contributes to resurgent currents. Interestingly, given a similar comment by Reviewer #2 (see above), we looked into the possibility that FGF14 might also be involved. Molecular data from a previous publication (Lovell et al., BMC Genomics, 2018) revealed low expression of this subunit in RA, and lack of differential expression with the surrounding arcopallium. Thus the expression of FGF14 is poorly correlated with the regional differences in resurgent currents we observed. We have now corrected our language regarding Navβ4 as a molecular requirement and we have an additional comment on FGF14. Please see lines 597-601.

While it seems that the resurgent current is closely related to the distinctive firing behavior of RAPN, especially the ability to fire repetitive action potentials, the contribution of other ion currents (other than the transient sodium current) which in turn influences spike shape and firing, was missing. Developmental changes may also occur to the entire complement of ion channels present in these neurons, thus contributing to the variations in the firing phenotype of RAPN seen through development. Such a study for example exploring pharmacological manipulations of RAPN is beyond the scope of the present paper, but as written these limitations are overlooked.

We also completely agree with the reviewer's comment and now comment on the possibility of other specific channels potentially contributing to changes in the AP waveform as an important theme for future studies, in lines 130-131 and 508-510.

In the Discussion the involvement of potassium channels to the repetitive and narrow spike firing phenotype is discussed, while the Results section imply that the resurgent current is solely promoting the narrow action potentials (lines 256-257). Indeed, presence of the resurgent current affects spike morphology, however, more studies are needed to prove that the resurgent current is solely responsible for this behavior. The Results section needs to be modified accordingly.

We appreciate the reviewer's comments, which reflect a similar comment from Reviewer #2. It was not our intention to imply that resurgent currents are the only mechanism underlying narrow action potentials. We have now additionally changed the text in the Results (line 130-131) and Discussion (lines 508-510).

lines 195-213 are confusing. What is the significance of examining the developmental regulation of subunit 1 given that (in contrast to subunit 4) its function is not known/not stated by the authors? The authors come to the conclusion at the end of this section that subunit 4 is correlated with the increase in resurgent sodium current throughout vocal learning development. What is then the role of subunit 1? Is it also developmentally regulated but by a different behavior? Can its role be omitted? What if the role of subunit 4 that is facilitation of high frequency firing seen in mammals does not apply in songbirds?

The reviewer raises here several related questions that require clarification. First, with regard to the last question, we would like to emphasize that the present data, as well as previous data in chicken as cited in the paper (Lewis and Raman, *J. Neurosci.*, 2011), provide substantial evidence linking Nav β 4 to resurgent currents and correlation with high frequency firing, thus supportive of conserved function for this subunit between birds and mammals. Next, while the present study is primarily focused on Nav β 4, we observed that Nav β 1 and Nav β 3, which are in the same family of modulatory subunits as Nav β 4, also undergo marked developmental switches in expression in RAPNs between 20 dph and adulthood. While we agree that less is known about the function of these other subunits, reporting the developmental regulation of Nav β 1 and 3 in birds seems worthwhile, for several reasons: (a) In mammals, these subunits share several molecular features, including binding to alpha subunits; (b) Nav β 1 and 3 undergo a similar developmental switch in mammals (Isom and Hull, *Neuropharmacology*, 2018), thus the developmental switch we have observed in finches seems to reflect a conserved regulatory trait shared by birds and mammals; (c) Nav β 3 shows high sequence conservation with mouse (79% residue identity and 87% conservation considering conservative substitutions); as for Nav β 1, while the available zebra finch sequence data is limited, data from a closely related songbird (Bengalese finch) with more complete sequence reveals 70% conservation (including conservative substitutions) with mouse, suggesting conserved function; (d) The predicted extracellular domains of Nav β 1, 3 and 4 in songbirds are all recognized as belonging to the Nav β family of modulatory subunits by predictive algorithms (InterPro Scan), indicative of conserved molecular structure with mammals; (e) There is evidence in mammals linking Nav β 1 to the modulation of Nav1.6 localization and/or gating kinetics (Zhao et al., *J. Neurophysiol.*, 2011; Brakenbury et al., *PNAS*, 2010). It seems to us worthwhile pointing to this possible conserved function, given the overall sequence conservation and what we observed in terms of activation and inactivation kinetics, but we also acknowledge that Nav β 1 function remains to be experimentally examined in birds.

We have now made changes to the Discussion, to make these points more explicit in the revised text. Please see lines 573-576 in the discussion and 797-804 in the Methods where we discuss this.

Minor:

Line 78: We provide “strong” evidence linking “extensive” might be better if you feel a need for a qualifier

Thank you for the suggestion. We have changed to “compelling” evidence.

Line 141: “These results reveal an exceptionally large INaR in RAPNs compared to those recorded in mammalian and chicken brainstem neurons”

provide some summary quantification here to help the reader

Thank you for the suggestion. We have provided summary quantification. Please see line 471-473.

Line 256: The persistent Na⁺ current mentioned is not referred to in the indicated figure (Fig. 5h).

We have now indicated the persistent sodium current in the figure legend. Figure 5 now has extra labels in several panels.

Line 434: “well suited” better than well "engineered"?

Thank you for the suggestion. The word “engineered” has been removed.

Line 480: Sentence starting “Taken together...” is confusing. The intrinsic properties of RAPNs differ from those of HVCPNs, but what about those intrinsic property differences represent a “starkly contrasting” difference? Also, since HVC is thought to be a site of temporal coding, how does the sentence make sense?

We acknowledge that discussion of comparisons between excitable properties of HVC and RA is beyond the scope of this manuscript and we have removed this sentence.

Line 496: spelling: “Deveopmental”

We have made the correction. We have done a spellcheck.

indicate IR-DIC in Supp 5 legend

We have added this text.

REVIEWER COMMENTS

Reviewer #1 (Remarks to the Author):

This revised version of "Resurgent Na⁺ currents promote ultrafast spiking in projection neurons that drive fine motor control" includes adequate action in response to major and minor comments from the previous version. Notably, the authors are to be commended for including some behavioral (song) data in the form of a new experiment.

The explanation of the new experiment, however, could be more accurately framed by taking into account data from a couple of labs that indicate that exposure to tutor song does not alter major metrics of singing behavior (e.g. age of initiation, rate), or RNA profiles in motor nuclei (likely because the males are singing, whether or not they have had tutor experience). Rather, a direct effect of tutor experience is found in non-motor brain areas of the song learning circuit and thus evidence converges on tutor experience influencing a critical period in learning that does not rely on motor nuclei such as RA. In this more integrated and updated framework, the results of this experiment are consistent with existing data, but not in the context of a tutor-dependent critical period per se.

As a corollary, the addition of the song-isolation experiment highlights an important distinction in the wording throughout the MS. The effects are often considered "age-dependent" effects but of course, males who differ in age also differ in their extent of singing experience. Both are relevant to patterns of gene expression generally, and as of yet, not experimentally dissected as factors here (these experiments are appropriate as future studies and non-essential at this stage). A simple wording and/or explanation change would remedy this simplification in the text, and make the overall story more cohesive now with the addition of the song-isolation experiment.

I suggest the methodological description for this experiment may be improved. "around" an age is arguably too vague for developmental work – what is the range of ages and how many animals fall into each day post-hatch? It could be a bit clearer: ~30dph, the males are placed all by themselves without the ability to hear, touch, or see any other bird, i.e. this is not simply song isolation but complete isolation? From what portion of the lights-on phase are the songs to be quantified collected and how far apart are they? Please clarify: for the SAP analysis, it was all of the experimental birds tested pairwise against each other? Or against all of the males used as tutors?

Minor points

Line 689: typo: previous*ly* described

Line 691: how was the relative intensity of the ISH DIG label standardized such that pixel intensity could be a validated range of numbers?

Line 826: might be helpful at this point to indicate this is from a normally-reared adult male

Line 835-ish (Figure 1): are these data all from the same bird/same RA slice?

Reviewer #2 (Remarks to the Author):

The authors have done a very nice job responding to the prior review. The new data comparing socially isolated and normally reared (tutored) zebra finches are interesting. The information on FGF14 expression is helpful. I have no additional concerns or suggestions.

Reviewer #3 (Remarks to the Author):

Authors have done a good job responding to the reviewer comments and revising the paper. The framing of the paper is also improved, and the potential connection to Betz cells and mammalian motor cortex will help to make these results of interest to a wider audience. This paper could also have provided some elaboration on coding issues as described in the first of the extracellular recordings from RA in singing birds, but its lacking is only a small limitation.

Dear Reviewer's,

We appreciate the positive evaluations and careful reading of the text and figures. We detail below (**in bold**) how we have addressed the Reviewer's specific suggestions and requests. All changes are highlighted (**in red**) in the revised manuscript.

Reviewer #1 (Remarks to the Author):

This revised version of "Resurgent Na⁺ currents promote ultrafast spiking in projection neurons that drive fine motor control" includes adequate action in response to major and minor comments from the previous version. Notably, the authors are to be commended for including some behavioral (song) data in the form of a new experiment.

We appreciate the Reviewer's positive comments.

The explanation of the new experiment, however, could be more accurately framed by taking into account data from a couple of labs that indicate that exposure to tutor song does not alter major metrics of singing behavior (e.g. age of initiation, rate), or RNA profiles in motor nuclei (likely because the males are singing, whether or not they have had tutor experience). Rather, a direct effect of tutor experience is found in non-motor brain areas of the song learning circuit and thus evidence converges on tutor experience influencing a critical period in learning that does not rely on motor nuclei such as RA. In this more integrated and updated framework, the results of this experiment are consistent with existing data, but not in the context of a tutor-dependent critical period per se.

This is an important comment that suggests a stronger framework for the bird isolation experiments. While the lack of exposure to a tutor is known to affect the timing of song crystallization (e.g. Morrison and Nottebohm, 1993), we agree that there is currently no evidence that exposure to a tutor has a major impact on general measures of singing behavior in juveniles (e.g. amount or rate of singing, or age of onset of sub-song; see new references: Mori and Wada, 2015, Ross et al, 2019), whereas even deafening does not alter the developmental progression of gene expression in RA (Mori and Wada, 2015). In contrast, there is evidence that developmental gene expression patterns in RA can be largely affected by singing behavior (Hayase et al, 2018). We also agree that there is considerable evidence that tutor exposure exerts significant impact on non-motor areas involved in song learning (e.g, Yanagihara and Yazaki-Sugiyama, 2016), thus many of the direct effects of tutor exposure are expected to occur at sites that precede RA. We therefore agree that RA function in vocal-motor output would not necessarily be expected to be directly influenced by tutor exposure, and have re-framed the rationale of this experiment accordingly. Nonetheless, we note that the motor commands received by RA during song production are likely distinct in tutored vs isolate birds, as the songs produced by these two groups are respectively the result of imitation of an external model and resemble the tutor song vs. the product of self-referencing vocal practice that does not resemble the tutor or the control sibling's song. Furthermore, tutor song exposure has been shown to affect the intrinsic properties of HVC neurons (Ross et al, 2019). In our study, despite the differences in song, the intrinsic excitable properties of RAPNs were indistinguishable between tutored and isolate siblings, thus appearing to be independent of tutor song exposure and imitation. We believe this is an important insight that helps inform and guide future experiments aimed at understanding factors that contribute to the development of intrinsic excitable features of RA. To address the Reviewer's comment, we have modified how we present the experiment (Results, lines 316-328), and

how we interpret its outcomes (Discussion, lines 543-551), and cited the relevant references accordingly.

As a corollary, the addition of the song-isolation experiment highlights an important distinction in the wording throughout the MS. The effects are often considered “age-dependent” effects but of course, males who differ in age also differ in their extent of singing experience. Both are relevant to patterns of gene expression generally, and as of yet, not experimentally dissected as factors here (these experiments are appropriate as future studies and non-essential at this stage). A simple wording and/or explanation change would remedy this simplification in the text, and make the overall story more cohesive now with the addition of the song-isolation experiment.

We fully agree with the Reviewer’s insightful comment that the term ‘age-dependent’ does not fully address the fact that older birds also had more opportunity to practice their song compared to younger ones. Thus an age comparison by itself has an intrinsic behavioral confound that our current study does not address. The isolate experiment thus provides initial insights into an important experiential factor, but does not fully dissect the confound of singing behavior. Following the Reviewer’s suggestion, we have opted to keep the term age-dependent but provide a statement about this important caveat in the relevant paragraph in the Discussion (lines 537-541). We note that the term age-dependent adequately reflects how the data were gathered without experimental manipulations of singing behavior. We thus believe that with our additional explanation of this important caveat, the term age-dependent would be acceptable and reasonable.

I suggest the methodological description for this experiment may be improved. “around” an age is arguably too vague for developmental work – what is the range of ages and how many animals fall into each day post-hatch? It could be a bit clearer: ~30dph, the males are placed all by themselves without the ability to hear, touch, or see any other bird, i.e. this is not simply song isolation but complete isolation? From what portion of the lights-on phase are the songs to be quantified collected and how far apart are they? Please clarify: for the SAP analysis, it was all of the experimental birds tested pairwise against each other? Or against all of the males used as tutors?

We thank the Reviewer for pointing to the need for further accuracy in our methodological description of the isolate experiment. To address this request, we now clarify the range of ages involved (namely ± 2 days around the target date). We also confirm that the birds were placed in complete isolation from other birds (20 dph: father/tutor removed; 30 dph: one bird removed and placed in isolation, and father returned; 65 dph: recordings and sacrifice). Thus, the study addresses not only tutor exposure but a complete lack of social interaction, the data showing that neither is required for normal development of RAPN excitable properties. The songs were consistently recorded during a three day period (63-65 dph) in the morning between lights-on (10 AM) and noon (12 PM). The songs of tutored birds were compared to the songs of their fathers (tutors). We did not quantify the acoustic similarity between isolates and their fathers since their songs were highly divergent upon qualitative assessment. For stereotypy, the analysis involved pairwise comparisons across multiple song renditions from each individual bird, not across individuals. A group comparison for the stereotypy index between tutored and isolated birds was done with a t-test. We have modified the related paragraph in Methods (lines 664-680) to provide the further details described above.

Minor points:

Line 689: typo: previous*ly* described

We have made the correction.

Line 691: how was the relative intensity of the ISH DIG label standardized such that pixel intensity could be a validated range of numbers?

We appreciate the reviewer's comment about pixel intensity measurements and now provide additional clarifications in the Methods on how we standardized our experimental conditions. Briefly, all slides for a given probe are run together using standardized and pre-established hybridization conditions (i.e. fixation time, wash times and temperatures, amount of probe, concentration of chromogen, duration of incubation in detection solution). These conditions result in brain sections with levels of non-specific background (density measurements taken over tissue regions lacking cells) that are essentially indistinguishable from hybridizations performed with no-probe controls. We also include a strongly constitutively expressed gene in our hybridizations (*gad2*) and control tightly the timing in chromogen to prevent signal saturation compared to this more highly expressed control gene. Lastly, since the density of RA cells per sq unit area changes with age, we normalized the module of the background-subtracted optical density values by the number of cells in RA, estimated on adjacent Nissl-stained sections. Under our optimized conditions, the densitometric values obtained were consistent and monotonic with labeled cell counts. We have used similar densitometric approaches in previous studies to quantify developmental gene expression in song nuclei of juvenile zebra finches (e.g. Lovell et al, 2011; Olson et al 2011) and/or regional differences in gene expression in the zebra finch arcopallium (Nevue et al, 2020). We have modified the related paragraph in Methods (lines 698-705 & 730-735) to provide the further details described above.

Line 826: might be helpful at this point to indicate this is from a normally-reared adult male
Done.

Line 835-ish (Figure 1): are these data all from the same bird/same RA slice?

These data come from multiple slices across 6 different adult male zebra finches. We have now clarified the number of birds used to obtain this data. See edited Figure 1 caption.

Reviewer #2 (Remarks to the Author):

The authors have done a very nice job responding to the prior review. The new data comparing socially isolated and normally reared (tutored) zebra finches are interesting. The information on FGF14 expression is helpful. I have no additional concerns or suggestions.

We thank the Reviewer for the positive evaluation of our effort.

Reviewer #3 (Remarks to the Author):

Authors have done a good job responding to the reviewer comments and revising the paper. The framing of the paper is also improved, and the potential connection to Betz cells and mammalian motor cortex will help to make these results of interest to a wider audience.

We thank the Reviewer for the positive evaluation of our effort, in particular the assessment of the relevance of the comparison with Betz cells and how it may help broaden the impact of our study.

This paper could also have provided some elaboration on coding issues as described in the first of the extracellular recordings from RA in singing birds, but its lacking is only a small limitation.

Following a previous reviewer recommendation, we have avoided detailed discussions of temporal coding issues, as addressing those would require data from *in vivo* recordings. We do cite multiple times a seminal paper on coding precision using extracellular recordings from RA in singing birds (Chi and Margoliash , 2001). In addition, we also cite now a recent paper on *in vivo* spike recordings from RA in singing birds (Yu and Margoliash, 1996).

REVIEWERS' COMMENTS

Reviewer #1 (Remarks to the Author):

I appreciate the author's thoughtful and thorough responses to previous comments. This revision addresses all major and minor concerns.